# Biomimetic Tumour Model Systems for Pancreatic Ductal Adenocarcinoma in Relation to Photodynamic Therapy

**DOI:** 10.3390/ijms26136388

**Published:** 2025-07-02

**Authors:** Olivia M. Smith, Nicole Lintern, Jiahao Tian, Bárbara M. Mesquita, Sabrina Oliveira, Veronika Vymetalkova, Jai Prakash, Andrew M. Smith, David G. Jayne, Michal Heger, Yazan S. Khaled

**Affiliations:** 1Leeds Institute of Medical Research, St. James’s University Teaching Hospital, Leeds LS9 7TF, UK; olivia74@hotmail.co.uk (O.M.S.); nicolemaylintern@icloud.com (N.L.); andrewmalvernsmith@me.com (A.M.S.); d.g.jayne@leeds.ac.uk (D.G.J.); 2Jiaxing Key Laboratory for Photonanomedicine and Experimental Therapeutics, Department of Pharmaceutics, College of Medicine, Jiaxing University, Jiaxing 314001, China; jiahaotian96@gmail.com; 3Department of Pharmaceutics, Utrecht Institute for Pharmaceutical Sciences, Utrecht University, 3584 CG Utrecht, The Netherlands; b.s.maiamesquita@uu.nl (B.M.M.); s.oliveira@uu.nl (S.O.); 4Department of Molecular Biology of Cancer, Institute of Experimental Medicine of the Czech Academy of Sciences, 142 00 Prague, Czech Republic; veronika.vymetalkova@iem.cas.cz; 5Engineered Therapeutics Group, Department of Advanced Organ Bioengineering and Therapeutics, Technical Medical Centre, University of Twente, 7522 NB Enschede, The Netherlands; j.prakash@utwente.nl; 6School of Medicine, The University of Leeds, LS2 9JT Leeds, UK; 7Membrane Biochemistry and Biophysics, Department of Chemistry, Faculty of Science, Utrecht University, 3584 CS Utrecht, The Netherlands

**Keywords:** pancreas, cancer models, preclinical testing, chemotherapy, photodynamic therapy, in vitro analysis, cell viability assays, cell death, 2-D and 3-D cell culture, spheroids and organoids, patient-derived xenografts

## Abstract

Pancreatic ductal adenocarcinoma (PDAC) is the most common type of pancreatic cancer and is associated with poor prognosis. Despite years of research and improvements in chemotherapy regimens, the 5-year survival rate of PDAC remains dismal. Therapies for PDAC often face resistance owing in large part to an extensive desmoplastic stromal matrix. Modelling PDAC ex vivo to investigate novel therapeutics is challenging due to the complex tumour microenvironment and its heterogeneity in native tumours. Development of novel therapies is needed to improve PDAC survival rates, for which disease models that recapitulate the tumour biology are expected to bear utility. This review focuses on the existing preclinical models for human PDAC and discusses advancements in tissue remodelling to guide translational PDAC research. Further emphasis is placed on photodynamic therapy (PDT) due to the ability of this treatment modality to not only directly kill cancer cells by minimally invasive means, but also to perturb the tumour microenvironment and elicit a post-therapeutic anti-tumour immune response. Accordingly, more complex preclinical models that feature multiple biologically relevant PDAC components are needed to develop translatable PDT regimens in a preclinical setting.

## 1. Introduction

Pancreatic ductal adenocarcinoma (PDAC) remains one of the most lethal solid cancers, accounting for approximately 9600 annual deaths in the United Kingdom [1,2]. Complete surgical resection remains the only curative option but is seldom achieved, with positive resection margins (R1) reported in up to 70% of cases [3,4,5]. With complete surgical resection, 5-year survival rates of ~40% are reported [6,7]. Approximately 70% of resected cases present with lymph node involvement, whilst distant metastases during surgical exploration are found in up to 25% of patients. Both are associated with a significantly decreased 5-year survival rate [8,9,10]. These statistics suggest that PDAC is a systemic disease and current therapeutics should focus on the unique biology of this cancer type.

Despite improvements in adjuvant and neoadjuvant chemotherapy regimens, chemotherapy approaches have had marginal benefits, with 5-year overall survival (OS) rates increasing from 6% to only 9% between 2014 and 2018 [11]. The observed chemoresistance is believed to be associated with inefficient drug penetration through the fibrotic stroma [12,13]. Therefore, there is a clinical need to create models representing the unique biological features of PDAC, and in particular the dense tumour microenvironment (TME) and its cellular and acellular constituents. These can be used to test novel chemotherapeutics [14] and immunotherapeutics [15] as well as experimental modalities such as photodynamic therapy (PDT) [16,17] to ultimately realise better therapeutic outcomes.

The PDAC TME is unique in that it comprises a thick desmoplastic stromal matrix composed of a variety of structural and cellular elements [17,18,19]. Among the most prominent are collagen, fibroblasts, and pancreatic stellate cells (PSCs) [20]. PDAC also bares a distinct immunosuppressive microenvironment owing to the presence of CD4^+^ T cells, cancer associated fibroblasts (CAFs), tumour-associated macrophages (TAMs), and myeloid derived suppressor cells (MDSCs) [21]. An anti-tumour immune response would usually require CD8^+^ effector T cells [22,23], though these are sparse in PDAC and usually deactivated [22]. Since the efficacy of PDAC immunotherapy relies on the existence of the patient’s anti-tumour immunity, responses are often poor [24]. Therapeutic recalcitrance is further exacerbated by the relatively low density of intratumoural microcirculation and vascular narrowing due to desmoplastic tissue-induced compression [25], affecting the delivery of systemically administered therapeutics. Although the latter phenomenon is more difficult to circumvent, studies are focusing on the reversal of immunosuppression to increase the efficacy of anti-PDAC agents. Extensive genomic, proteomic, and molecular analyses have expanded our understanding of the heterogeneity of PDAC and disease progression [1,26,27]. However, modelling PDAC in a research setting remains challenging due to the biological heterogeneity—a key factor in the development of personalised and precision therapies—that is arduous to emulate.

The aim of this literature review is to summarise and critically appraise the state of the art regarding PDAC models, focusing on spheroids, hybrid culture models, organoids, scaffold-based models, assembloids, microfluidic models, and xenografts derived from cell lines, organoids, and patients. This information is presented as a backdrop to PDT research on PDAC inasmuch as PDT is directly cytotoxic to tumour cells whilst harnessing the ability to damage the TME and trigger an anti-tumour immune response—i.e., factors that otherwise dictate therapeutic recalcitrance.

## 2. Two-Dimensional and Three-Dimensional PDAC Models

The biology of various human PDAC cell lines has been extensively studied [28]. Two-dimensional (2-D) in vitro models, which comprise monolayers of cells in culture, play an important role in the development of anti-PDAC drugs and have added to our understanding of how PDAC cells develop, proliferate, invade, and respond to anti-cancer drugs [29,30,31,32]. These models have been instrumental in medical research and are low-cost, easy to create, and facilitate fast data curation [33]. Large quantities of cells can be grown, which also enables high-throughput screening [34].

Though these models have proven useful, they are associated with numerous limitations. Primarily, a 2-D model cannot faithfully represent the complex PDAC TME [33]. Extracellular protein expression, interactions between the extracellular matrix (ECM) components and cells, tumour heterogenicity, and the dense, fibrotic stroma that houses non-parenchymal cells are not truly represented [33,35]. Most PDAC cell lines are derived from rapidly growing PDAC tumours, so slower growing tumours are underrepresented [36]. Furthermore, Audero et al. [37] demonstrated the biological significance of acidic stress in the TME. Tumour acidification is a well-established phenomenon that results from tumour cells switching to anaerobic respiration as a result of hypoxia (Warburg effect). The authors demonstrated that exposure to acidic conditions selects PDAC cells with augmented migration and invasion abilities induced by epithelial–mesenchymal transition, potentiating their metastatic potential when re-exposed to a neutral pH. The rewiring occurs through transcriptomic changes that affect genes responsible for proliferation, migration, epithelial–mesenchymal transition, and invasion. Naturally, 2-D cell cultures are always kept at near-neutral pH, which obviates these acid-mediated processes that likely materialize in the more complex models (e.g., Figure 1).

Advancements in in vitro models of PDAC have led to three-dimensional (3-D) model types that better represent the PDAC TME [38], which is crucial for the proper appraisal of anti-PDAC therapeutics [39]. Future studies should ideally utilise the following models to build more robust systems and increase the success rate of translational research.

### 2.1. Spheroid Models

Modelling the complex TME and 3-D structure of PDAC accurately has been challenging. Technological developments have allowed the creation of more advanced PDAC models from tumours using 3-D models that contain or are contained within artificial matrices. Spheroid models are clusters of cells within a 3-D structure. These models can be sourced from cell lines, tumour cells, or tumour tissues [40]. Cells are grown in a low-adhesion microplate to promote spheroid formation and must be cultured with growth factors and preferably ECM as well as components of the PDAC stroma, including CAFs [33,41]. Their 3-D organisation consists of cell–cell aggregation and proliferating heterogenous, necrotic, and quiescent cells within layers [42]. The methods available to culture spheroids have been reviewed elsewhere [43,44].

Proliferation-, metabolic-, and pH gradients can be represented by spheroid models [33,45]. This was shown in a study that investigated whether the mechanosensitive ion channel Piezo1 plays a role in transducing mechanical signals using PSC spheroids [46]. Acidification of the intracellular space inhibited Piezo1-mediated Ca^2+^ influx into PSCs in PDAC spheroids [46]. Ware et al. attempted to create a 3-D in vitro PDAC spheroid model to help increase our understanding of stroma development and tumour–stroma interactions [47]. Spheroid models made with PANC-1 cells and PSCs were structurally more compact and proliferated more than spheroids without PSCs. Researchers have also been able to model cell–ECM and cell–cell interactions present in PDAC using spheroids [48]. ECM components such as collagen I and III, smooth muscle actin (SMA), and fibronectin (a cell adhesion protein) have been successfully incorporated into PDAC spheroids and confirmed by immunohistochemistry [47]. Spheroids can also be used in high-throughput systems and for investigating the toxicity of novel PDAC drugs. Dufau et al. used Capan-2 pancreatic spheroids to evaluate spatio-temporal dynamics of chemotherapeutics. The authors compared the toxicity of gemcitabine alone and in combination with the checkpoint kinase-1 (CHK1) inhibitor 4-[[(3S)-1-azabicyclo[2.2.2]octan-3-yl]amino]-6-chloro-3-(1,3-dihydrobenzimidazol-2-ylidene)quinolin-2-one (CHIR-124) [49]. When gemcitabine was combined with CHIR-124, further cytotoxicity was induced in Capan-2 spheroids (~68% less ATP content was found, indicating lower cell viability) than with gemcitabine alone [49].

Many studies have suggested that spheroids are able to model drug resistance [40,50,51]. One study treated PANC-1, PSC, BXPC3, and Capan-1 pancreatic spheroids and corresponding 2-D models of these cell lines with a microtubule inhibitor, CB13, which inhibits cell division [40]. Significantly more chemoresistance occurred in the 3-D models than in the 2-D models following treatment. A decrease of ~59% in cell viability occurred in the 2-D models vs. 3% in 3-D models of the BXPC3 cell line [40]. It is believed that the chemoresistance exhibited by spheroids is due to the difficulty of chemotherapeutics in penetrating the multi-layered spheroid structure (Figure 1).

#### Limitations of Spheroid Models

The use of spheroids for PDAC modelling has several limitations. This model does not completely replicate the TME. For example, mono-culture spheroids do not model cellular diversity and cell–cell interactions and also lack the immune infiltrate which the PDAC tumour evades as well as the dense matrix and stroma [35,52,53]. Spheroids have low reproducibility, can be easily broken or removed during pipetting, and should be carefully handled [45]. Evidence also suggests that spheroids made using cell lines cannot accurately represent apicobasal polarity that plays an important role in cell migration [54,55]. As a result, spheroids are not ideal for modelling the physiological changes that occur in PDAC.

### 2.2. Co-Culture Models

Co-culture models for PDAC attempt to recapitulate the TME by incorporating multiple cell types, such as CAFs, immune cells, and endothelial cells. Around 90% of the volume of PDAC tumours are made up of supportive tissues as well as TME components [56]. The supportive tissues in PDAC tumours consist of pancreatic connective tissue, lymphatic vasculature, infiltrative immune cells, CAFs, and stroma [57,58,59,60]. The role of CAFs is to help regulate cytokines, growth factors, immune filtrate, and ECM deposition in the TME. These cells comprise a substantial proportion of the stroma and are the largest contributor to collagen, proteoglycan, and hyaluronan production [61,62]. CAFs are highly heterogeneous and exist within distinct subpopulations; each play a unique role in the pathogenesis of pancreatic cancer, so the incorporation of these different cell types into PDAC models is essential. A 2021 study developed a mouse model that mimicked the functional and histological characteristics of CAFs in PDAC [63]. The authors co-transplanted adipose-derived mesenchymal stem cells (which helped to generate CAFs) and the PDAC Capan-1 cell line into mice [63]. Global RNA sequencing, histological analysis, and single cell-based RNA sequencing showed that myofibroblast (myCAFs), inflammatory (iCAFs), and antigen presenting (apCAFs) CAF subtypes were represented in the model [63]. Accordingly, models of PDAC can be improved by co-culturing cancer cells with components such as stellate cells, fibroblasts, or other stromal cells [64,65,66,67] as shown in Figure 1.

Evidence suggests that co-culturing spheroids can represent a higher level of drug resistance due to the ability to model a dense PDAC stroma that drugs have difficulty penetrating [66]. One study used PANC-1 spheroids that were either mono-cultured or co-cultured with PSCs to compare gemcitabine resistance [66]. Cell viability was 5% greater in the co-cultured vs. mono-cultured spheroids [66]. Drug resistance was also greater in the co-cultures, though this was not significant. However, this study was limited to investigating one type of chemotherapeutic; the difference may be more apparent with other types of drugs. Another study compared the mechanical stiffness of PANC-1 monocultured spheroids and spheroids co-cultured with PSCs and transforming growth factor beta-1 (TGF-β1) [68]. PANC-1 co-cultured spheroids supplemented with TGF-β1 could model significantly more mechanical stiffness than the monocultures [68]. For example, the complex shear modulus (indicator of mechanical stiffness) increased by ~93% following 45 days of culture [68]. This may explain why co-cultured models are better for representing drug resistance in PDAC.

Evidence further suggests that co-culturing other 3-D models with CAFs can induce heterogeneity [57]. One study showed that organoids (another type of 3-D in vitro model) co-cultured with CAFs induced differential expression of SMA [55]. SMA plays an important role in PDAC progression and metastasis by suppressing E-cadherin, a tumour suppressor protein that prevents cell dissociation [69,70,71]. Öhlund et al. investigated whether subtypes of CAFs with distinct phenotypes exist in PDAC organoids co-cultured with PSCs, which are precursors of CAFs [72]. They found two CAF subpopulations; one, myCAFs, showed elevated expression of α-SMA and produced desmoplastic stroma. The other secreted interleukin 6 (IL-6) and other inflammatory mediators (iCAFs). This study shows that co-cultured models may be useful in helping to personalise treatment strategies by employing distinct cellular phenotypes in light of the fact that patients have different CAF subtype populations.

Co-culture models have been proficiently used for testing anti-PDAC drugs, drug delivery systems [73], and biomolecule-targeted interventions [74]. Kuninty et al. observed that microRNA (miR)-199a-3p and miR-214-3p were induced in 2-D cultures of patient-derived pancreatic CAFs and TGF-β-activated human PSCs, and that inhibition of miR-199a and miR-214 using hairpin inhibitors blocked TGFβ-induced differentiation markers (collagen, α-SMA, platelet-derived growth factor β receptor), migration, and proliferation. Using a heterotypic spheroid model comprised of human PDAC (PANC-1) cells and human PSCs it was demonstrated that spheroid size was smaller when human PSCs were transfected with anti-miR-199a and anti-miR-214 compared to control anti-miR [32]. Anane-Adjei et al. investigated whether hyperbranched polymers would offer a suitable drug delivery system for PDAC [75]. They also assessed whether 2-D cell culture or MIA PaCa-2 spheroids co-cultured with bone marrow-derived mesenchymal stem cells (BM-MSCs) could better mimic the stromal tissue in PDAC since a small population of these cells can be observed in pancreatic tumours [75,76]. MSCs are adult stem cells that can differentiate into different cell types such as adipocytes (fat cells), osteocytes (bone cells), or chondrocytes (cartilage) [77]. Therefore, the authors compared the toxicity of conjugated hyperbranched N-(2-hydroxypropyl)methacrylamide (HPMA)-gemcitabine polymers and free gemcitabine in the mono-culture vs. co-culture spheroids. Higher cell viability was seen following treatment with 100 μM of free gemcitabine in the co-culture spheroids (~67%) compared to the mono-culture spheroids (~50%), demonstrating a higher level of drug resistance in the co-cultures [75]. However, no significant difference in drug resistance was seen between the co-culture and mono-culture spheroids when treated with HPMA polymers.

#### Limitations of Co-Culture Models

Co-culture models represent valiant efforts to maximally encapsulate the compositional features of the PDAC parenchyma and TME. Nevertheless, co-culture models often lack full tumour heterogeneity and the different genetic and epigenetic alterations in cells that comprise the PDAC in that most co-culture models include only a few selected cell types. Moreover, many co-culture models do not fully mimic the stiffness, composition, and mechanical properties of the actual tumour stroma [78]. While some co-culture models incorporate immune cells (e.g., macrophages or T cells), the models often lack the full repertoire of tumour-infiltrating lymphocytes, MDSCs, and dendritic cells. In that respect, an immune-suppressive environment of PDAC is difficult to recreate in vitro. PDAC tumours exhibit abnormal, hypoxic vasculature and corollary alterations in metabolism, which influences cancer progression and drug delivery. Most co-culture systems lack functional blood vessels, leading to unrealistic oxygen and nutrient gradients. Aside from the fact that many co-culture models fail to mimic the chemoresistance observed in vivo, drug diffusion in 3-D cultures rarely matches in vivo pharmacokinetics. Moreover, in vitro models do not fully replicate the metabolic interactions between cancer and stromal cells that include lactate recycling [79,80] and glutamine dependence [79,81]. The complexity of the system gives rise to additional technical challenges, such as those observed during long-term culture where co-culture systems degrade over time due to differences in cell proliferation rates. Cancer cells can outgrow other cell types with lower proliferation rates, leading to imbalances in the model. Finally, as with any model, variability in cell sources, passage numbers, and culture conditions leads to inconsistent results across studies. The lack of standardised protocols renders comparisons between different models difficult.

### 2.3. Organoid Models

Another type of 3-D in vitro models are organoids. These models are distinct from spheroids in that they are made using mechanically or enzymatically dissociated tumour samples as well as 2-D cell lines, whilst spheroids are only created from 2-D cell lines [45]. The organoid tissue is usually maintained in a Matrigel or collagen scaffold, in a growth factor-enriched medium that provides nutrients for tumour growth, or in suspension such as air-liquid interface culture (containing medium and collagen) [82]. The sources of these models include tumour cells, patient-derived resected tumour tissues, and embryonic, adult, and pluripotent stem cells (stem cells that have been genetically reprogrammed into embryonic stem cells (ESCs) and are able to differentiate into any cell type) [33,83,84]. Using patient-derived resected tumour tissues to make organoids is particularly useful as it helps to reflect different CAF subtypes and specific characteristics in PDAC, including subclone epigenetic, phenotypic, and metabolic diversity. Their complex composition, as presented in Figure 2, consist of many cell lineages and helps to model the structure and function of PDAC. Self-assembled differentiated cells that are responsive to physical and chemical cues form part of the 3-D structure [85,86,87]. Laboratories experienced in generating organoids using surgically resected tumour tissue or endoscopic biopsies have had an organoid formation success rate of 75–95% [88].

Organoid models have enabled intercellular and intracellular interactions, the cellular compartment, diversity, composition, and structure of PDAC to be modelled ex vivo [33,45,89]. They also have allowed many aspects of PDAC to be modelled without requiring an in vivo system, such as proteomic, histological, and genetic features and self-renewal [90]. Driehuis et al. demonstrated many comparable characteristics between 30 patient-derived organoid (PDO) lines from tumours in the pancreas and their corresponding primary tumours [91]. For example, a patient showing progressive disease under gemcitabine treatment had a corresponding PDO that was the most resistant to gemcitabine out of all of the organoids [91]. Three other patients with stable disease following gemcitabine treatment and their corresponding PDOs showed high or intermediate sensitivity [91]. When sequenced, PDAC organoids exhibited many similarities to PDAC tumours. One study assessed whether any similarity could be observed between PDAC organoid models and human PDAC by sequencing miRNA profiles of extracellular vesicles (EVs), a promising diagnostic tool for PDAC [92]. Interestingly, the same set of miRNA EVs could be observed in both PDAC organoids and in blood plasma samples of PDAC patients. Organoids can also be passaged continuously for ongoing experimentation [93].

Organoid models usually consist of many different cell types. The cells occupy niches and therefore enable the modelling of the interactions and physiological characteristics observed in the native tumour [45]. Holokai et al. attempted to develop a clinically relevant PDAC organoid model that could be used to predict the efficacy of targeted therapeutics [94]. Their pancreatic organoids modelled the stromal and immune components seen in PDAC, including myCAFs and tumour-infiltrating lymphocytes. Depletion of arginase 1-expressing polymorphonuclear MDSCs, which block CD8^+^ T cell anti-tumour immune responses rendered the organoids susceptible to anti-programmed death 1 receptor (PD-1)-programmed death ligand 1 (PD-L1)-induced death [94]. The data demonstrate that PDAC organoids are suitable for predicting the efficacy of targeted therapeutics.

Since the median time to create PDOs and pharmacotyping can be less than 2 months, depending on the tumour quantity and quality, one potential use for this model could be to assist a patient in their post-surgical recovery period to select adjuvant therapy [91,95]. Driehuis et al. attempted to describe a biobank of pancreatic PDOs characterised by methods such as DNA sequencing to show the importance of personalised medicine predicated on organoids. Different pancreatic organoid lines had distinct drug sensitivity profiles that could be sorted for 76 different anti-cancer agents (microtubule-targeting drugs, AURKA-targeting drugs, PIK3CA-targeting drugs, and TOP1-targeting drugs) [91]. PDAC organoid models could also be used to investigate disease physiology and the targeting of specific signalling pathways by novel therapeutics. For example, Krieger et al. recently tried to determine whether transcriptionally and histologically defined subpopulations, which had classical features in metastases, were associated with more aggressive clinical behaviour in PDAC [96]. They used PDAC organoids to identify classical subtypes of genes such as p081, which correlated to better chemotherapy response, including to gemcitabine.

#### Limitations of Organoid Models

Organoid models have several limitations. They can take longer to make and have higher costs than spheroid models, and pH gradients are unable to be modelled accurately [45,97]. In addition, PDOs derived from those receiving neoadjuvant therapy is dependent on viable tissue being present during resection [98]. Comparing data can also be challenging since Matrigel and collagen have interbatch differences in composition [35]. For example, Matrigel includes other components as well as collagen, including laminin, which promote the invasion phenotype in tumour cells [99,100].

### 2.4. Hydrogel Scaffold-Based Models

Hydrogel scaffold-based models comprise covalently bound hydrophilic polymers arranged in a 3-D network that retain a large amount of water [101,102]. The structure of the cells within the network influences cell function and responses to PDAC therapeutics [102]. Traditional hydrogel scaffold-based models can be made using gelatin methacrylate or Matrigel hydrogel.

Technological advancements have led to the development of hydrogel scaffold-based models that consist of a cellular network structure offering impressive biochemical, biocompatible, and biophysical tunability [102]. These models help promote 3-D cell proliferation, replicate the ECM, and allow for the diffusion of nutrients [102] (Figure 3). Recently, this was illustrated in a study where hyaluronic acid- and gelatin-based hydrogels were used to create matrices for PDAC spheroids comprised of ASPC-1 cells and CAFs to model the PDAC TME [103,104]. Higher amounts of vascular endothelial growth factor receptor 2 (VEGFR2) were observed in stiffer hydrogels compared to softer hydrogels. This receptor mediates angiogenesis and subsequent progression of PDAC by allowing cells to obtain oxygen and nutrients from leaky blood vessels [103]. Other advantages of hydrogel scaffold-based models are that these models can represent both soft and elastic characteristics due to the presence of networks of hydrophilic polymers and the ability to swell in aqueous solution [105]. Curvello et al. assessed the ability of collagen-nanocellulose hydrogels to mimic the PDAC ECM [106]. When type I collagen fibrils and cellulose nanofibers were blended, the resulting hydrogel scaffold exhibited controllable stiffness and modelled the natural PDAC tissue.

Hydrogel scaffold-based models can be combined with other model types for PDAC. Spheroids can be cultured with hydrogels to model stronger cell–cell interactions and recreate the mechanical forces seen in vivo [107]. Ermis et al. created a hydrogel-based spheroid PDAC model comprising CAFs and ASPC-1 cells to more closely recapitulate the desmoplastic stroma [103]. They demonstrated very compact tissue formation with increased matrix stiffness compared to models without these cellular constituents [103].

#### Limitations of Hydrogel Scaffold-Based Models

The use of hydrogel scaffold-based models requires consideration. For one, the models lack spatial and temporal control, which leads to an uneven distribution and irregular seeding of cells [108,109]. The complex microvasculature seen in PDAC is absent in these models, resulting in a deficiency in nutrient and signalling molecule transport [108]. As a result, seeded cell viability and function are lost relatively quickly over time. This may limit the range of anti-PDAC drug toxicological investigations.

### 2.5. Assembloid Models

An assembloid model consists of multiple organoid types that self-organise into 3-D cell-based systems [110]. Assembloid models can be made with different tissues or cells, as shown in Figure 4. Formed using diverse cell lineages, the structures can facilitate the physiological changes seen in PDAC, enable cell–cell interactions, and demonstrate the effects of interventions on different organs.

Choi et al. investigated the role of jagged-1 in PDAC cell plasticity since plasticity is poorly understood in organoids yet may influence drug efficacy [111]. Assembloids were made using peripheral blood mononuclear cells (PBMCs), endothelial cells, and CD44(−) cells isolated from PDAC organoids using fluorescent activated cell sorting (FACS). To investigate jagged-1 expression in pancreatic cancer compared to healthy pancreatic tissue, the authors analysed the Cancer Genome Atlas, the Pancreatic Adenocarcinoma dataset and the Genotype-Tissue Expression dataset [111]. Jagged-1 expression was significantly increased in malignant tissue. They also analysed jagged-1 levels in the assembloids during differentiation, which revealed an increase in jagged-1(+)CD24(+)CD44(-)epithelial cell adhesion molecule (EpCAM)(+) cells in the assembloids. Though these models have demonstrated their potential in representing the various cellular and structural features observed in PDAC, this is a new technology and further studies and investigations will evolve this technique.

#### Limitations of Assembloid Models

While assembloids provide a more physiologically relevant system than the abovementioned, more traditional models, the assembloid models also come with several limitations. Because of their intricate nature, these models are plagued with complexity and standardisation issues. For example, assembloids require precise control over cellular composition, ratios, and spatial organisation, making these models technically demanding to generate. Differences in cell sources (e.g., patient-derived vs. cell line-derived) can lead to variability in reproducibility, especially in the face of lacking universal protocols. As was addressed for co-cultures, assembloids may experience loss of architecture over time and suffer from differential growth rates of individual cellular constituents, leading to loss of specific cell populations. Due to their 3-D nature and dense structure, insufficient oxygen and nutrient diffusion can lead to central necrosis, limiting viability. Moreover, desmoplastic stroma may not be fully recapitulated and immune cell–TME interactions are commonly incomplete (see also Section “Limitations of Co-Culture Models”). Assembloids require advanced culture conditions (e.g., hydrogels, bioreactors) and rely on expensive reagents (e.g., Matrigel, growth factors, specialised media) that limit high-throughput screening while increasing cost. The previously addressed issues related to vascularisation and consequential hurdles for drug permeability and pharmacokinetics (Section “Limitations of Co-Culture Models”) also apply to assembloids. While patient-derived assembloids, which require fresh tissue samples that can be limited and pose ethical restraints, retain genetic heterogeneity, this also leads to variability in responses, complicating broad therapeutic predictions. Despite their complexity, these models do not fully replicate all in vivo conditions, as a result of which results may not always translate to the clinical setting.

### 2.6. Microfluidic Models

Microfluidic devices are miniaturised systems comprised of chambers and micro-scale fluidic circuits that mimic the PDAC TME by incorporating fluid flow, multiple cell types, and controlled biochemical gradients (Figure 5) [112]. The devices’ conceptual framework was designed to overcome the limitations of earlier in vitro models of PDAC, including difficulties in modelling heterogeneity and biochemical gradients. PDAC spheroids can be cultured within the device, which facilitates the use of fewer cells and reagents due to its microscale dimension. This allows precise and quantitative measurements to be performed [112,113]. Microfluidic models are low-cost and easy to use since only small volumes of samples are needed [114]. Therefore, the model is suitable for high-throughput screening and labour-intensive work.

Microfluidic models manufactured with human or murine cells embedded in 3-D ECM may also be used to feature in vivo behaviour such as the generation of shear stress during extravasation [112] and epithelial-mesenchymal transition and local invasion [115]. The geometry of blood vessels and the 3-D PDAC TME can also be demonstrated. One study investigated the effect of shear stress on the morphology and behaviour of human umbilical vein endothelial cells (HUVECs) using a microfluidic device [116]. Various levels of uniform wall shear stress were demonstrated using a novel microfluidic shear stress generator. Future studies could use microfluidic systems to control biophysical and biochemical conditions in the model more precisely [112]. In addition, multifluidic models can be adapted to form multi-organ-on-chip models. As this technology develops, it would be particularly useful in PDAC research since it can facilitate and monitor cross-communication between organs, thereby modelling systemic disease states in relation to PDAC. It is apparent that microfluidic models can be developed in numerous ways to create an advanced PDAC ex vivo model.

Microfluidic models would be useful for industrial work as well as large-scale PDAC therapeutic testing. In one study, a novel microfluidic model was manufactured using a cyclin olefin polymer chamber named HepaChip and cultured with PDAC PANC-1 spheroids [117]. This model was able to withstand higher doses of cisplatin when compared to other in vitro cultures.

#### Limitations of Microfluidics Models

Microfluidic devices require precise engineering and components (pumps, imaging systems, and precise fluid control) that makes design, fabrication, and assembly relatively complex and costly. Operating microfluidic systems requires specialised knowledge in microengineering and biofabrication, which in turn limits accessibility. Accessibility is further hampered by the fact that many PDAC-on-chip models are custom-built and not widely commercially available. Even small variations in chip fabrication can affect reproducibility, making cross-study comparisons challenging. Each chip must be carefully assembled and experiments often require continuous monitoring, making these models tedious. Consequently, microfluidic devices are generally low-throughput and difficult to use for large-scale drug screening, also owing to the possibility of only testing a few conditions at a time (unlike high-throughput spheroid or organoid systems). Drug testing is additionally complicated by a lacking liver metabolism and systemic pharmacokinetics infrastructure, limiting drug response accuracy. While microfluidics can model drug flow, the systems do not fully replicate the diffusion barriers seen in dense PDAC tumours whilst excessive or inadequate shear stress may alter drug effects in ways not seen in actual tumours.

Aside from these technical limitations there are also biological hurdles. First, current microfluidic models struggle to fully replicate ECM stiffness, composition, and cellular heterogeneity while long-term immune cell-tumour interactions are difficult to sustain in microfluidic systems despite the possibility to introduce immune cells. Although cancer, stromal, and endothelial cells can be incorporated, current models often lack certain key cell types, such as pericytes, MDSCs, and dendritic cells. Second, many microfluidic cultures cannot be maintained long-term due to nutrient depletion, fluid shear stress, or loss of cell viability and, unlike organoids or assembloids, these systems may fail to maintain proper 3-D tissue architecture over time. Third, current microfluidics models suffer from vascularisation and perfusion challenges. While some models incorporate artificial vasculature, true capillary networks with functional endothelial barriers are difficult to recreate. Emulating hypoxic PDAC conditions and interactions between PDAC cells and circulating immune cells or tumour-derived exosomes is hence arduous. As alluded to previously for other models, there is no universal protocol for PDAC microfluidic models, which can produce interstudy inconsistencies in these models too.

### 2.7. Xenograft Models

In a xenograft model, tissue or cells are transplanted into an animal (generally subcutaneously or orthotopically) to represent PDAC and emulate drug responsiveness more closely [118,119]. Different types of xenograft models include cell line-derived xenografts, organoid-based xenografts, and patient-derived tumour xenografts (PDX). The tissues or cells are usually implanted into an immunodeficient [120] or syngeneic mouse (where tumour cells are obtained from genetically identical animals) [121]. This avoids graft rejection by the host’s immune system (Figure 6).

#### 2.7.1. Cell Line-Derived Tumour Xenograft Models

Cell line-derived xenograft models consist of human PDAC cell lines transplanted into immunocompromised mice [120]. These models improve the representation of the biology of PDAC and the responsiveness to drugs compared to the aforementioned models [36]. The models are practically more convenient than PDX models since no patient tissue samples are exacted. Cell line-derived xenograft models may be better alternatives to PDX models in instances where an improved in vivo model is desired but time is limited or patient access is restricted since no patient tissue needs to be obtained. Future studies could also incorporate the chicken embryo chorioallantoic membrane (CAM) model into the cell line-derived xenograft model line up since these are able to assist cell growth. Rovithi et al. cultured four primary PDAC cell lines (PDAC-1, PDAC-2, PDAC-3, and PDAC-4) on CAMs to generate low passage cell cultures and transduced the cell lines with lentivirus expressing firefly-luciferase (Fluc) [122]. By incorporating bioluminescence, the authors could measure tumour growth in situ. As cell numbers increased, Fluc activity increased. 

##### Limitations of Xenograft Models

It has been reported that cell line-derived xenografts have predicted the therapeutic response in various cancers, including PDAC, inconsistently [123,124,125]. Bruns et al. created a pancreatic orthotopic xenograft model to investigate whether the anti-epidermal growth factor receptor (EGFR) antibody cetuximab inhibits PDAC growth and metastasis and whether gemcitabine exacerbates this effect [126]. The model was further used to examine therapeutics targeting EGFR as a potential treatment for PDAC. The authors observed that there was a reduction in tumour volume with cetuximab compared to the untreated control group. Moreover, the reduction in tumour volume was more profound when treatment was combined with gemcitabine (~94.7% decrease in tumour cells for 32 ng/mL gemcitabine plus 2.5 μg/mL of cetuximab vs. a decrease of 10.5% in tumour cells without gemcitabine and only 2.5 μg/mL of cetuximab [126]). In contrast, it was found in a phase III clinical trial that patients who had advanced pancreatic cancer treated with gemcitabine and a placebo or gemcitabine with cetuximab experienced no benefit from treatment [127]. Other studies also further support the translational inconsistency between results seen with cell line-derived xenografts and clinical trials for various types of cancers [128,129]. One reason for the differences seen in drug responses may be due to the fact that cell line-derived xenograft models exhibit limited stromal infiltration, absent interactions with the adaptive immune system, and grow mainly as homogeneous masses of tumour cells [36]. However, issues with a lack of stromal infiltration in cell line-derived tumour xenograft models may be resolved by co-culturing these models with stromal components such as stellate cells, thereby modelling the thick desmoplastic stromal matrix in PDAC and drug resistance, albeit not overcoming inherent immune system defects.

#### 2.7.2. Organoid-Based Xenograft Models

Organoid-based xenograft models (also referred to as patient-derived organoid xenografts) can be generated by transplanting PDAC tumour organoids into an immunocompromised mouse [82]. Organoid-based xenograft models represent tumour neoplastic cell heterogeneity and can emulate the different stages of PDAC disease progression [36], which may be due to the recovery of different stem cells during transplantation that reflect these varying stages [36].

Xenografts made using organoid models can recapitulate all stages of PDAC disease progression, making them distinct from other xenograft models. In addition, transplantation into a host is the only efficient way of incorporating blood vessels into organoids [130] that are also amenable to representative locoregional stressors such as vascular compression [131]. Interestingly, one study created a patient-derived PDAC organoid xenograft model that showed intraepithelial neoplasms progressing in an indolent or invasive way and associated these growth patterns with PDAC subtypes (classical or basal-like) [132]. Moreover, organoid-based xenografts help to recapitulate the large amounts of collagen seen in PDAC that is not represented in cell-based xenografts [133,134]. Recently, Tanaka et al. attempted to create a well-established preclinical model of PDAC to test new therapeutic targets [135]. They created a PDAC organoid-based xenograft model using the S2-013 cell line and performed pathological and immunohistochemical analysis to characterise this model. The model had similar tissue to that seen in PDAC patients, with abundant cancer stroma containing mature blood vessels and collagen [135]. As the vasculature and TME can be represented by this model, its use could be applied in the search for novel therapeutic targets in PDAC, including anti-angiogenics.

Another suitable use for organoid-based xenograft models would be to investigate personalised therapy for PDAC patients. Raimondi et al. attempted to investigate the feasibility of using patient-derived PDAC organoid xenograft models to screen for the response of oncolytic adenoviruses as personalised therapy [136]. They found that differences were seen in cytotoxicity with the oncolytic adenoviruses in different patient-derived PDAC organoid xenograft models, indicating the sensitivity to oncolytic adenoviruses seen in primary PDAC tumours [136]. The cell viability in an organoid derived from a PDAC patient following treatment with 1 × 10^4^ pfu/well of the oncolytic adenovirus AduNuPARmE1A was ~27 ± 12% compared to ~83 ± 10% in an organoid derived from another PDAC patient [136].

##### Limitations of Organoid-Based Xenograft Models

Although organoid-based xenograft models fulfil an important role by bridging the gap between in vitro organoid cultures and traditional PDXs, the models do come with several limitations. Because of their complexity, these models require more resources that include immunodeficient mice, specialised facilities, and long-term maintenance, which drives up cost. Organoids take weeks to expand in culture and xenografts require weeks to months to develop into mature tumours, delaying experimental timelines. As a result, organoid-based xenograft models have low throughput, which makes these models impractical for large-scale drug screening. Logistical hurdles also come in the form of ethical concerns regarding animal experimentation and the use of immunodeficient mice in preclinical research. These are compounded by the possibility that organoid-based xenograft models may face challenges in regulatory approval for clinical drug validation, as they are not fully humanised. Mice process drugs differently from humans [137] and the xenografts therefore do not fully replicate human drug absorption, distribution, metabolism, excretion, and toxicity (ADMET), which may lead to differences in pharmacological profiles and limit direct clinical translation.

From a biological perspective, the murine host provides stromal components that are of non-human origin, which may not fully mimic human CAFs, ECM, and desmoplastic responses. Inasmuch as immunocompromised mice lack key components of adaptive immunity, which is critical in tumour control and eradication [138], no accurate studies on immune evasion and immunotherapy are possible. Furthermore, organoid-derived tumours may not fully recapitulate the dense fibrosis in PDACs, leading to altered drug penetration and response compared to human tumours. Organoids are typically expanded in vitro before implantation, which can lead to clonal selection and loss of intratumoural heterogeneity. Serial passaging of organoids before implantation may introduce genetic and epigenetic alterations [139,140] and certain subpopulations of stem-like or therapy-resistant cells may be lost during the organoid culture phase [141,142], making the xenograft model less representative of the original tumour. Another important biological factor that is often dismissed is local biochemical milieu. Most organoid-based xenograft models involve subcutaneous implantation, which may neither properly reflect the molecular landscape in the pancreas (e.g., abundance of digestive enzymes (trypsin, amylase, lipase) and hormones (insulin, glucagon)) nor accommodate natural tumour initiation and metastatic dissemination.

#### 2.7.3. Patient-Derived Xenograft Model

PDXs are a type of xenograft model that use tumour tissue engraftment from patients transplanted into mice [143]. The direct sourcing of patient tumour tissue has allowed PDX models to improve in vivo modelling by representing the complex PDAC TME with higher accuracy. PDX models may also be employed to identify unique therapy sensitivities in precision medicine [144]. Recently, Magouliotis et al. attempted to investigate the suitability of PDAC PDX models for precision therapy [145]. The xenografts modelled the most and least aggressively differentiated population of the patient’s PDAC tumours using immunohistochemistry [145]. This investigation confirmed that the models are suitable for the exploration of precision medicine for PDAC [145]. Evidence also suggests that these models may be most suitable for assaying the efficacy and efficiency of PDAC therapeutics [35]. Wu et al. utilised PDX PDAC models to test the feasibility of gemcitabine-based nanoparticles as a potential treatment for PDAC [146]. The nanoparticles could inhibit tumour progression and alleviate systemic toxicity. In a different study, Garcia et al. studied the anti-tumour efficacy of JQ1, a bromodomain protein inhibitor, in PDAC PDX models [147]. They found that JQ1 inhibited the growth of all the PDAC PDX models tested [147].

##### Limitations of Patient-Derived Xenograft Models

PDX models have issues associated with their use that echo organoid-based xenograft models. These models require high quantities of tissue and it can be difficult to resect enough tissue required for studies within a specific timescale. The models also usually require a high number of animals and take time to grow [35,148,149], providing a timeline that is often unrealistic [35] and costly [36]. These factors may contribute to the lower success rates of de novo PDX models compared to cell line-established models [35]. Another limitation of these models is their inability to accurately represent interactions with immune components in the TME due to the use of immunodeficient mice [150]. Therefore, this model cannot be used for immunotherapy research [151] unless the model employs humanised mice [152]. Passaging PDX models also results in clonal evolution, meaning that this model should only be used within three generations for therapeutic experiments [153]. Therefore, it may be difficult for large-scale serial passaging and expansion to be completed for the purpose of drug screening [36].

A summary of each different model type described in this review article is summarised in Table 1.

### 2.8. Evolving Technologies for PDAC Mimicry

Though existing models of PDAC have improved substantially and can now replicate the TME quite accurately, they are complex to develop and maintain and translatability remains a major challenge. An alternative approach would be to adopt an in silico strategy. This often involves computational models that comprise databases, molecular modelling approaches, machine learning, data mining, and/or data analysis tools [154]. The use of in silico systems is associated with lower costs, better research ethics, and facilitates rapid data collection. As a result, utilisation of technology in this way will be particularly useful as it advances, particularly in the era of artificial and generative intelligence.

In silico models have many uses within research and may be particularly useful in the investigation of novel therapeutics for PDAC. Recently, one study used datasets available on the Gene Expression Omnibus database to investigate potential PDAC biomarker targets [155]. Based on their co-expression and protein–protein interaction networks, 79 gene candidates were enumerated. Five significant endoplasmic reticulum protein processing pathways involved in PDAC progression were found, including hsa04141 [155]. Having previously been associated with malignant behaviour, another study analysed transportome (membrane transporter and channel) expression changes in PDAC and their correlation with functional and behavioural responses [156]. The authors filtered data from a DNA microarray Affymetrix GeneChip dataset using cut-off fold-change values of ≤2 or ≥2. Among several notable observations, some down-regulated genes, including calcium voltage-gated channel subunit alpha1 G, were linked to cell differentiation. These observational studies exemplify the use of in silico techniques for the investigation of PDAC biology and druggable targets. Follow-up confirmation in biological systems is always warranted after in silico analysis.

## 3. Application of Biomimetic PDAC Models in PDT Research

PDT is a relatively new approach that aims to overcome the recalcitrance to systemic therapy exhibited by many types of cancer, including PDAC [157]. PDT has been approved for the treatment of superficially located lesions and solid tumours [158,159] and is under investigation in clinical trials with respect to tumours in internal organs [160,161]. In the case of the latter cancer types, direct cytotoxic effects are conferred via the photo-excitation of a systemically administered photosensitiser (PS) and subsequent generation of reactive oxygen species [162] in the target tissue (Figure 7). The reactive transients chemically modify vital molecules (e.g., proteins [163,164], lipids [165], nucleic acids [166]), resulting in excessive and often irreparable damage that entails loss of membrane integrity [167], perturbation of cellular homeostasis [168], metabolic catastrophe [169], and eventually the manifestation of various modes of tumour cell death [170,171] and immunological cell death [172].

The application of PDT for PDAC is particularly relevant for several reasons. At mildly elevated levels, reactive oxygen species (ROS) are needed to steer tumour progression through cell transformation [173], proliferation [174], survival [175], angiogenesis [176], and metastasis [177,178]. As a protective measure, cancer cells upregulate anti-oxidant defence systems to maintain a sustainable level of redox homeostasis [179]. Hyperoxidative stress, as is caused by PDT (and other therapeutic modalities) [180,181], is a prelude to impending cell demise that can be offset by an anti-oxidant response to PDT [158,169,182]. These variables notwithstanding and as discussed in Section 3.3.3, PDT affects intratumoural redox states in a manner that is conducive to a positive treatment response beyond direct photochemical damage. PDT can destroy both the PDAC cells and surrounding desmoplastic tissue [17] whilst eliciting a post-therapeutic anti-tumour immune response [183] and inducing metabolic paralysis of parenchymal and non-parenchymal cells owing to vascular shutdown [184,185]. In addition to perturbing the tumour-promoting effects of PDAC stroma, including tumour–stroma interactions, PDT can enhance the response to chemotherapy [186] and immunotherapy [187]. Therefore, the use of models that faithfully represent and encompass the complex biology and biochemistry of PDAC are crucial in understanding and optimising PDT procedures.

To date, studies on PDT have incorporated a variety of preclinical PDAC models outlined above and detailed below.

### 3.1. In Vivo PDAC Models Used in PDT Research

#### Mouse Models

Animal models of human PDAC are currently the most utilised for PDT-related research on PDAC as these models embody the sum of a PS’s pharmacokinetics, pharmacodynamics, toxicology, and disposition (ADMET). Initial PDAC studies used chemically induced xenograft models, though these were quickly replaced by cell line alternatives due to inefficiency [188,189]. The PDAC cells obtained via the chemical route were directly injected into the animal, usually orthotopically [190]. Whilst the obtained tumours retained some of the characteristics of PDAC, human cell lines (Table 2) are preferred in PDT studies nowadays as these better represent the clinically observed histopathological features [191]. The tumour xenografts also reflect the reaction of the PDAC TME to therapy, which include post-PDT anti-tumour responses [192,193]. Immunological responses are best studied in syngeneic xenograft models featuring orthotopic inoculation. Vascular infiltration into the xenografts not only provides the necessary conduits for PS delivery but allows studying the effects of intratumoural vascular shutdown following PDT [194], which gives rise to tumour hypoxia [195,196] and activation of specific survival pathways in residually viable tumour cells [158,182,197]. Survival signalling may have deleterious consequences on therapeutic outcome and can be combatted through adjuvant routes [158]. However, the PDAC stroma and its heterogeneity cannot be modelled precisely in homotypic cell-based xenografts as genetic and phenotypic features are restricted in immortalised cell lines [198].

To partly remediate this issue, a mixture of tumour and stromal cells can be incorporated into the inoculation bolus to produce heterotypic xenografts. One study co-implanted MIA PaCa-2 and human pancreatic cancer-associated fibroblast (pCAF) cells into mice. PDT plus vitamin D3 receptor activation in fibroblasts led to a reduction in tumourigenic signalling [199]. Another study using the same approach confirmed that a high degree of desmoplasia can be achieved in PDAC + pCAF xenografts [200]. PDT-treated pCAF-replete PDAC tumours exhibited substantial tumour necrosis and a 1.5-fold reduction in collagen density. PDT-induced collagen destruction was associated with better progression-free and overall survival [201], attesting to the importance of PDAC stroma disruption in treatment efficacy.

Although the hybrid approach enables the study of PDT effects on particular aspects of the stroma, the model cannot fully represent actual disease heterogeneity when single-source cell lines are used. As alluded to before, PDAC tumours bear different stromal subtypes with variations in collagen content, immune cell subsets, endothelial cells, and CAF populations [202]. Ideally this variation is reflected in the model as it is associated with different therapeutic responses, including to PDT [203,204].

### 3.2. Ex Vivo PDAC Models Used in PDT Research

#### PDX Models

PDX or organoid-based xenograft models entail patient-derived tissues comprising native stromal components and cell types and therefore properly resolve the structural and cellular heterogeneity dilemma pertaining to the PDAC TME. To date, PDXs have not been employed in PDT/PDAC research. Nevertheless, science can to an extent be borrowed from other cancer types, such as a recent study that used a bladder cancer PDX model to investigate a novel type of chemo-PDT [205]. Histological analysis revealed that the histopathological features of the parental tumour, including cell and tissue structures, were retained in the PDX after transplantation into mice. The uptake and distribution of the nanoformulated PS as well as the tumour response to PDT were characterised and included changes in the level of proteins related to apoptosis, DNA damage, and cytoskeletal aggregation. These findings are particularly useful given that PDX responses are believed to be correlated to the responses observed in patients during clinical treatment [206].

Next-generation PDAC models that fully recapitulate the TME are still under development. PDX models could incorporate patient-derived tumour cells as well as adjacent normal tissues cultured together with patient-derived stem cells [207], which may in part dictate the tumour’s susceptibility to treatment [208]. PDAC PDX models developed in this way would better reflect the complex histological and genetic attributes of the comprehensive tumour milieu. PDXs would further serve as a tool to model PDAC progression, which is pertinent in that PDAC responds to therapy according to the stage of development and molecular landscape [203]. In addition, though PDT clinical trials have focused on locally advanced PDAC, studying PDT for later-stage PDAC is warranted [17] given the potency of abscopal effects [209,210]. Accordingly, PDAC PDX models could facilitate the development of more personalised PDT modalities and, if indeed successful, augment the rate of clinical translation of preclinical research findings as well as promote and expand the use of PDT for a greater number of PDAC patients.

### 3.3. In Vitro PDAC Models Used in PDT Research

#### 3.3.1. Cell Culture Monolayers (2-D)

Two-dimensional cell cultures are well suited for the investigation of novel PSs, light sources, dosages, and molecular mechanisms as they offer a standardised, reproducible system into which virtually all molecular and cell biology techniques can be plugged. These models often serve as a validatory preface to follow-up in vivo studies. For example, a recent study employed four PDAC cell lines (Capan-1, Capan-2, MIA PaCa-2, and PANC-1) to evaluate the targeting and therapeutic effectiveness of a novel folate-conjugated PS to its cognate receptor folate receptor 1 (FOLR1) [211]. FOLR1 expression was confirmed using qRT-PCR and cells were analysed for PS uptake and localisation, dark toxicity and phototoxicity, and post-PDT immune signalling (cytokine secretion and peripheral blood mononuclear cell activation). The positive in vitro data were subsequently replicated in a humanised SCID mouse model of human PDAC in terms of short-term tumour destruction (complete removal within 9 days after PDT), validating the utility of the in vitro data.

Moreover, genetic modifications through transduction, transfection, or editing (e.g., CRISPR/Cas) has enabled studies on targeting, ligand binding, endocytic mechanisms, and signal transduction pathways using overexpressed or specifically expressed proteins and site-directed pathway modifications (mutagenesis) [212]. For instance, genetically modified cell lines have been utilised to identify proteins involved in the resistance of PDAC to PDT. This is relevant in light of the fact that PDAC signalling pathways have been linked to therapeutic recalcitrance [199,200]. In specific instances, 2-D cell cultures are sufficient in representing the effects of PDT on individual aspects of the PDAC stroma. For example, a 2023 study investigated fibroblast activation protein (FAP)-targeted PDT using 2-D layers of NIH-3T3 cells (mouse embryonic fibroblasts) transfected with FAP [213]. The model was able to demonstrate the in vitro binding and cytotoxicity of the treatment.

Despite the above-referenced benefits, 2-D models are associated with several shortcomings that curtail their utility. Most importantly, the models cannot mimic the native TME and do not account for the crosstalk and intercellular interactions observed in PDAC [214]. This may lead to non-translatable results, particularly in cases where PDT is used as a means to disrupt the TME [215].

#### 3.3.2. Heterotypic Spheroid Cultures

The preclinical investigation of PDT for PDAC has been accelerated by the use of 3-D models that are considered to have greater physiological proximity and predictive value in regard to therapeutic responses. As alluded to previously, a surrounding ECM is required to steer structural and intercellular interactions observed in the PDAC TME [216]. Desmoplasia and interstitial fibrosis in PDACs hamper the delivery of PSs and chemotherapeutics to the tumour parenchyma, resulting in reduced sensitivity to treatment in an immuno-friendly environment. Albeit complex outside of an in vivo environment, these factors should ideally be accounted for in the available in vitro models, especially in light of the increasingly pervasive RRR (reduce, reuse, and recycle) principle in animal research [217].

PDT investigations are commonly performed on PDAC spheroids that are co-cultured with CAFs. Accordingly, Saad et al. generated 3-D MIA PaCa-2 spheroids co-cultured with a varying percentage of PDAC-derived CAFs [218]. The model was employed to study the distribution of a PS-loaded, cancer cell-targeted photoimmunoconjugate (PIC) using fluorescent proteins as reporters. A key finding was that PICs could penetrate spheroids despite high levels of desmoplasia. Another study utilised spheroids composed of MIA PaCa-2, ASPC-1, or Capan-2 cells in combination with pCAFs to investigate the effect of low-dose PDT plus radiation therapy [219]. The spheroids treated by the combinatorial modality exhibited reduced growth and cell–cell adhesion and more profound necrosis and loss of integrity compared to the individual treatments.

Apart from the potentially negative implications on PSs delivery, fibroblasts have also been reported to secrete hepatocyte growth factor (HGF) to activate the HGF–MET signalling axis in a paracrine manner in ASPC-1 and MIA PaCa-2 spheroids [220]. Furthermore, c-MET controls cancer cell proliferation, survival, motility and invasion that, when dysregulated by anomalous c-MET activation, can lead to tumour growth and metastatic progression of cancer cells. When ASPC-1 and MIA PaCa-2 PDAC spheroids were co-cultured with human embryonic lung fibroblasts (MRC-5), the spheroids exhibited less susceptibility to benzoporphyrin derivative (BPD)-PDT compared to fibroblast-lacking spheroids. Although MET expression in ASPC-1 cells is inherently higher than MIA PaCa-2, the combination treatment of PDT and MET inhibition was equally effective in both spheroid test systems, especially at low radiant exposures (0.5–10 J/cm^2^).

#### 3.3.3. Hydrogel Scaffold-Supported Spheroids

Instead of using cellular ‘production factories’ (i.e., fibroblasts) for TME reconstruction, spheroids can be cultured in hydrogel scaffolds to imitate the PDAC TME. The robustness of the spheroid model increases when co-cultured with CAFs. Inasmuch as most PDT research is performed on heterotypic spheroid cultures, the hydrogel materials used for scaffolding of heterotypic spheroids are addressed next.

##### Matrigel

Matrigel is the commercial name for growth factor-replete, solubilised basement membrane matrix secreted by Engelbreth–Holm–Swarm mouse sarcoma cells that compositionally resembles the extracellular environment found in many tissues and is commonly used to culture cells (2-D and 3-D) [221,222]. It is the most widely used scaffold material in PDT studies that employ spheroids.

An example of seminal research obtained with PDAC/CAF co-cultures supported by a Matrigel scaffold was a study by the Hasan group [200]. As backdrop, therapeutic recalcitrance arising from fibroblast activity in the ECM has traditionally been ascribed to increased ECM formation, metabolic reprogramming, and heterotypic cell–cell interactions [223]. However, Broekgaarden et al. [200] furnished a redox-based explanation that encompasses multiple cellular components in spheroids. Cancer cells typically have inherently higher levels of ROS production due to aberrant cell growth-driven metabolic demand [224]. Elevated intracellular pro-oxidant states in cancer cells have been linked to resistance to therapy [225], but cancer cells do not seem to be solely responsible for the therapeutic recalcitrance in spheroids. The authors demonstrated that CAFs and, to a lesser extent, healthy dermal fibroblasts (HDF1) in Matrigel-supported MIA PaCa-2 spheroids increased redox states, as evidenced by increased cyclooxygenase-2 (COX-2, upregulated via nuclear factor kappa-light-chain-enhancer of activated B cells (NF-κB) in response to ROS [226]) and heme oxygenase 1 (HO-1, oxidative stress response protein [227]) protein expression. These phenomena, which were reproducible in vivo, concurred with resistance to BPD-PDT as well as oxaliplatin chemotherapy, and were dependent on the cell line combinations used. Less pronounced resistance to PDT and chemotherapy was observed for MIA PaCa-2 spheroids cocultured with HDF1 and especially CAF6 cells and was absent in ASPC-1/CAF6 spheroids. Additional factors that influenced treatment resistance were fibroblast activation status, spheroid size, and therapeutic dosage. It was further shown that metformin, a mitochondrial complex I inhibitor that blocks the passage of electrons along the electron transport chain during aerobic respiration [228], reduced oxidative stress in MIA PaCa-2/CAF microtumours without affecting cell viability. Since redox stress increased in CAF-lacking spheroids, metabolic rerouting for tumour sustenance was clearly mediated by CAFs through as yet unidentified mechanisms. Perturbation of CAF-mediated metabolic rerouting by metformin moreover was associated with increased therapeutic efficacy in MIA PaCa-2/CAF6 microtumours, although this result was again dependent on the PDAC and fibroblast cell lines used. Corroborative results in terms of redox states and their effect on PDAC metabolism and therapeutic susceptibility were obtained with rotenone, another mitochondrial complex I inhibitor [229], in PDT-subjected heterotypic organoid co-cultures as well as MIA PaCa-2 xenografts [230]. Taken together, the study demonstrated that these hybrid PDAC microtumours were able to recapitulate some of the essential elements of the PDAC TME, including cell–cell interactions, redox states, metabolic plasticity, and resistance to treatment, which allowed for more representative investigations of potential mechanisms of treatment escape and pharmacological interventions.

Although an excellent substrate for cell development and growth per se, there is lot-to-lot variability of numerous constituents in Matrigel. This variability has been shown to affect PDT and oxaliplatin therapeutic responses and incidentally exacerbate inter-spheroid size variability [200,231]. Moreover, batch differences may also affect spheroid survival before treatment [200]. The exact cause of the heterogeneity is currently unclear.

##### Collagen

As a more consistent alternative to Matrigel, collagen has proven to be useful scaffold material [78]. PANC-1 spheroids utilising collagen scaffolds exhibited greater invasiveness than when cultured in Matrigel. BPD-PDT treatment exerted more profound photocytotoxicity in an ECM-invading PANC-1 subpopulation of cells, especially in the leading cells that extended beyond a 200-μm radial distance from the spheroid’s edge but failed to restrict primary spheroid growth. This was in contrast to the notable cytostatic effect by oxaliplatin on the primary spheroid [232]. The study also used riboflavin-mediated photocrosslinked collagen hydrogels to investigate the growth and invasiveness of primary spheroids as a function of crosslink density [233], which is analogous to ECM density. This model allows control of hydrogel stiffness by regulating the degree of crosslinks through light dose. The invading velocity of the organoid-derived cells monotonically declined with increasing crosslink density. Furthermore, the cells were more sensitive to PDT than to oxaliplatin at lower crosslink densities. Unfortunately, the study did not clarify whether there is a relation between crosslinking degree and PDT response. This would have added valuable information in the sense that increased ECM stiffness (i.e., higher crosslink density) is expected to be inversely proportional to ease of PS penetration into the spheroid.

##### Alginate and Gelatin

Other materials for creating hydrogel matrices are alginate and gelatin, which are inexpensive and similar alternatives to the commercial ECM hydrogels used for spheroid culture. These hydrogels are compositionally versatile in that hydrogel stiffness can be regulated by adjusting the ratio of alginate or gelatin to water [231]. It is noteworthy that the purity of alginate should be maximal, as impurities increase the risk of residual endotoxins activating undesirable signalling (via e.g., CD14) that in turn may result in inter-batch spheroid response heterogeneity [234].

With respect to research results with alginate or gelatin-scaffolded spheroids, which at present are relatively scarce in the context of PDT, cationic liposomes encapsulating BPD were more avidly taken up by PANC-1 spheroids generated in alginate than anionic liposomes. The uptake levels were positively correlated with spheroid size [231]. In human breast cancer (MCF-7) 3-D bioprinted hydrogel-based spheroids treated with Ce6-PDT, apoptosis occurred at the top and bottom regions of single spheroids regardless of the vertical axis orientation of the light source and spheroid size [235]. In a co-culture of Matrigel-based PDAC spheroids and MRC-5 fibroblasts, a significant increase in IL-1α/α-SMA ratio was observed in the PANC-1/MRC-5 spheroids compared to PANC-1 spheroids co-cultured with PSCs [16,204]. Increased α-SMA-positive myCAFs subpopulations are normally considered to suppress tumour progression, while increased IL-1α expression can activate the generation of iCAFs and promote tumour behaviour [236]. Also, higher oxaliplatin resistance and BPD-PDT sensitivity occurred in the PDAC/MRC-5 spheroids compared to homotypic PDAC spheroids, regardless of which type of fibroblasts was used [204].

Taken together, these superimposed effects generated by the introduction of CAFs and the inherent differences of PDAC cell lines are particularly important for the assessment of PDT response [200,220,230].

**Table 2 ijms-26-06388-t002:** Non-exhaustive summary of human PDAC cell line-based models used in PDT research.

Cell Line	Disease	Source	Models	Methods	Tested in PDT	Ref.
**A818-1**	PDAC	Metastatic	Spheroids	plates coated with agarose in non-supplemented medium at a ratio of 1:3	No	[237]
**A818-4**	PDAC	Metastatic	Spheroids	nonadherent round-bottom plates with medium containing 20% methyl cellulose	No	[238]
**A818-6**	PDAC	Metastatic	Spheroids	plates coated with agarose or cultured in rotating culture vessels	No	[239]
**A** **S** **PC-1**	PDAC	Metastatic	Spheroids	ultra-low attachment round-bottom plates; ***PDT regimen***: BPD, 690 nm, 150 mW/cm^2^, 0–80 J/cm^2^	**Yes**	[219,220]
			Co-cultured spheroids	MRC-5, patient-derived CAFs; ***PDT regimen***: BPD, 690 nm, 150 mW/cm^2^, 0–80 J/cm^2^	**Yes**	[219,220]
			Microfluidic spheroids	spheroids were generated by liquid overlay method and then transferred to a microfluidic chip	No	[240]
			Hydrogel-based spheroids	PEG hydrogel	No	[241]
			Organoids ^†^	Matrigel; Cultrex Reduced Growth Factor BME, low attachment plates	No	[242,243]
			Cell line-derived xenografts	male nude mice (16 wk), subcutaneous; ***PDT regimen***: zinc phthalocyanine-loaded mesoporous silica nanoparticles, 685 nm, 50 mW/cm^2^, 100 J/cm^2^, 1980 s	**Yes**	[244]
			Cell line-derived xenografts	male SCID nude mice (6 wk), orthotopic; ***PDT regimen*:** verteporfin, 690 nm, 74 mW/cm^2^, 10–40 J/cm^2^, 135–540 s	**Yes**	[245]
**B** **X** **PC-3**	PDAC	Primary	Spheroids	medium containing 0.24% methylcellulose	No	[40]
			Co-cultured spheroids	MRC-5, suspended in polyacrylamide hydrogel coated with collagen type I	No	[246]
			Microfluidic spheroids	HepaChip device	No	[117]
			Hydrogel-based spheroids	Matrigel, collagen I; ***PDT regimen***: BPD, 690 nm, 100 mW/cm^2^, 0.5–25 J/cm^2^	**Yes**	[232]
			Organoids ^†^	Matrigel, collagen I, tumour-associated PSCs and M2 macrophages in suspension	No	[247]
			Cell line-derived xenografts	female athymic NCR-Nu-F nude mice (5–8 wk), subcutaneous	No	[248]
			Cell line-derived xenografts	BALB/c nude mice (6 wk), orthotopic; ***PDT regimen***: Ce6, 660 nm, 200 mW/cm^2^, 200 J/cm^2^, 1000 s	**Yes**	[249]
**Capan-1**	PDAC	Metastatic	Spheroids	medium containing 0.24% methylcellulose	No	[40]
			Co-cultured spheroids	PSCs	No	[250]
			Hydrogel-based spheroids	Matrigel and medium mixture (1:2)	No	[250]
			Cell line-derived xenografts	female BALB/c nude mice (6 wk), subcutaneous; ***PDT regimen***: rBC2-IR700, NIR light 670–710 nm, 100 J/cm^2^	**Yes**	[251]
			Cell line-derived xenografts	female BALB/c nude mice (6 wk), orthotopic; ***PDT regimen***: rBC2-IR700, NIR light 670–710 nm, 100 J/cm^2^	**Yes**	[251]
**Capan-2**	PDAC	Primary	Spheroids	ultra-low attachment round-bottom plates; ***PDT regimen***: BPD, 690 nm, 150 mW/cm^2^, 0.5–40 J/cm^2^	**Yes**	[219,252]
			Co-cultured spheroids	patient-derived CAFs; ***PDT regimen***: BPD, 690 nm, 150 mW/cm^2^, 0.5–40 J/cm^2^	**Yes**	[219]
			Cell line-derived xenografts	athymic nude mice, subcutaneous; ***PDT regimen***: temoporfin, 980 nm, 500 mW/cm^2^, 90 J/cm^2^, 180 s	**Yes**	[253]
**CFPAC-1**	PDAC	Metastatic	Spheroids	cancer stem cell medium	No	[254]
			Co-cultured spheroids	MRC-5 or PSCs; ***PDT regimen***: BPD, 690 nm, 100 mW/cm^2^, 0.5–25 J/cm^2^	**Yes**	[204]
			Organoids ^†^	collagen I, CFPAC-1 cells expressing GRHL2	No	[255]
			Cell line-derived xenografts	athymic CD1 nude mice (6–8 wk), subcutaneous	No	[256]
			Cell line-derived xenografts	female BALB/c nude mice (5 wk), orthotopic	No	[257]
**COLO 357**	PASC	Metastatic	Spheroids	nonadherent round-bottom plates with medium containing 20% methyl cellulose	No	[238]
			Co-cultured spheroids	patient-derived CAFs	No	[258]
			Hydrogel-based spheroids	gelatin-norbornene (GelNB)-based hydrogels	No	[258]
			Cell line-derived xenografts	female SCID/bg mice (4 wk), subcutaneous	No	[259]
			Cell line-derived xenografts	female SCID/bg mice (4 wk), orthotopic	No	[259]
**DAN-G**	PAC	Primary	Spheroids	polystyrene-coated ultra-low attachment plates	No	[260]
			Co-cultured spheroids	fibroblast-conditioned medium	No	[260]
			Cell line-derived xenografts	male NMRI nude mice (4–6 wk), subcutaneous	No	[261]
**HPAC**	PAC	Primary	Spheroids	round-bottom plates pretreated with 0.5% polyHEMA, plates coated with 1% agarose in DMEM; ***PDT regimen***: Ru-bqp-ester, 470 nm, 2.4 ± 0.2 mW/cm^2^, 4.3 ± 0.4 J/cm^2^, 1800 s	**Yes**	[262,263]
			Cell line-derived xenografts	female athymic BALB/c nude mice (7 wk), subcutaneous	No	[264]
			Cell line-derived xenografts	female SCID nude mice (5 wk), orthotopic	No	[265]
**HPAF-II**	PDAC	Metastatic	Spheroids	plates coated with 1% agarose in DMEM	No	[263]
			Co-cultured spheroids	fibroblast (DF-1) cells	No	[266]
			Cell line-derived xenografts	female athymic NCR-Nu-F nude mice (5–8 wk), subcutaneous	No	[248]
			Cell line-derived xenografts	female NOD/SCID nude mice (8 wk), orthotopic	No	[267]
**Hs 766T**	PAC	Metastatic	Hydrogel-based spheroids	Matrigel	No	[268]
			Cell line-derived xenografts	female athymic nude mice (6 wk), subcutaneous	No	[268]
**JoPaCa-1**	PDAC	Primary	Cell line-derived xenografts	NOD.Cg-Prkdcscid Il2rgtm1Wjl (NOD/SCID/c or NSG) mice, orthotopic	No	[269]
**KCI-MOH1**	PAC	Primary	Cell line-derived xenografts	female SCID mice (4 wk), subcutaneous	No	[270]
**KLM-1**	PDAC	Metastatic	Spheroids	NanoCulture plates	No	[271]
			Hydrogel-based spheroids	2-methoxyethyl methacrylate and 2-(diethylamino)ethyl methacrylate heteropolymer	No	[272]
			Cell line-derived xenografts	female athymic nude mice (5 wk), subcutaneous	No	[273]
			Cell line-derived xenografts	female NGS mice (5–6 wk), orthotopic	No	[274]
**KP-1N**	PASC	Metastatic	Cell line-derived xenografts	nude mice (6–8 wk), subcutaneous	No	[275]
**KP-2**	PA	Primary	Cell line-derived xenografts	nude mice (6–8 wk), subcutaneous	No	[275]
**KP-3**	PDAC	Metastatic	Cell line-derived xenografts	nude mice (6–8 wk), subcutaneous	No	[275]
**KP-4**	PA	Metastatic	Spheroids	ultra-low attachment round-bottom plates	No	[276]
			Cell line-derived xenografts	BALB/c nude mice (6–12 wk), subcutaneous	No	[277]
**MIA PaCa-2**	PDAC	Primary	Spheroids	ultra-low attachment round-bottom plates; ***PDT regimen***: BPD (including antibody-targeted BPD and liposomal BPD), 690 nm, 150 mW/cm^2^, 0–80 J/cm^2^	**Yes**	[200,218,219,220]
			Co-cultured spheroids	MRC-5, patient-derived CAFs; ***PDT regimen***: BPD (including antibody-targeted BPD and liposomal BPD), 690 nm, 150 mW/cm^2^, 0–80 J/cm^2^	**Yes**	[218,219,220]
			Microfluidic spheroids	HepaChip device	No	[117]
			Hydrogel-based spheroids	Matrigel; ***PDT regimen***: BPD, 690 nm, 150 mW/cm^2^, 1–50 J/cm^2^	**Yes**	[200,230]
			Organoids ^†^	Matrigel, collagen I, tumour-associated PSCs and M2 macrophages in suspension		[247]
			Cell line-derived xenografts	female nude mice (6 wk), subcutaneous; ***PDT regimen***: LC-Dox-PoP, 665 nm, 150 mW/cm^2^, 50 J/cm^2^	**Yes**	[278]
			Cell line-derived xenografts	male Swiss nude mice (4 wk), orthotopic; male Swiss nude mice (4–6 wk) co-implanted with pCAFs, orthotopic; ***PDT regimen***: BPD, 690 nm, 100 mW/cm^2^, 50 J/cm^2^; verteporfin or liposomal irinotecan, 690 nm, 100 mW/cm^2^, 75 J/cm^2^	**Yes**	[199,230]
**MZ-PC-1**	PDAC	Metastatic	Cell line-derived xenografts	NMRI nude mice (4–6 wk), subcutaneous	No	[279]
**PaCa-44**	PDAC	Primary	Cell line-derived xenografts	C.B-17/IcrHsd-Prkcdscid Lystbg mice (8–10 wk), subcutaneous	No	[280]
			Cell line-derived xenografts	C.B-17/IcrHsd-Prkcdscid Lystbg mice (8–10 wk), orthotopic	No	[280]
**PaCa 5061**	PDAC	Primary	Cell line-derived xenografts	male and female C57BL/6 mice (14–16 wk), subcutaneous	No	[281]
**Pan2M**	PDAC	Metastatic	Cell line-derived xenografts	female BALB/c nude mice (4 wk), orthotopic	No	[282]
**P** **ANC** **03.27**	PAC	Primary	Cell line-derived xenografts	athymic C57BL/6 nude mice, subcutaneous	No	[283]
**P** **ANC** **04.03**	PDAC	Primary	Co-cultured spheroids	PSCs	No	[284]
			Hydrogel-based spheroids	Matrigel and collagen I mixture (3:1)	No	[284]
**PANC 04.14**	PAC	Unknown	Cell line-derived xenografts	nude mice, orthotopic	No	[285]
**PANC 10.05**	PDAC	Primary	Cell line-derived xenografts	male nude mice (8 wk), subcutaneous	No	[286]
**PANC** **-1**	PDAC	Primary	Spheroids	Nunclon Sphera plates, NanoCulture plates; ***PDT regimen***: 6-amine-2,5-bromophenalenone (OE19), 525 nm, 18.6 mW/cm^2^, 16.6 J/cm^2^, 900 s	**Yes**	[271,287]
			Co-cultured spheroids	MRC-5, PSCs; ***PDT regimen***: BPD, 690 nm, 100 mW/cm^2^, 0.5–25 J/cm^2^	**Yes**	[204]
			Microfluidic spheroids	HepaChip device	No	[117]
			Hydrogel-based spheroids	Matrigel, collagen I, riboflavin-mediated collagen photocrosslinking hydrogel, alginate-gelatin hydrogel; ***PDT regimen***: BPD, 690 nm, 100 mW/cm^2^, 0.5–25 J/cm^2^; BPD, 690 nm, 150 mW/cm^2^	**Yes**	[204,231,232,233]
			Organoids ^†^	Matrigel, collagen I, tumour-associated PSCs and M2-like differentiated macrophages in suspension	No	[247]
			Cell line-derived xenografts	female BALB/c nude mice (6 wk), subcutaneous; female athymic CD1 mice (4 wk), subcutaneous (both 2D and spheroids-based); ***PDT regimen***: YLG-1, 650 nm, 100 J/cm^2^	**Yes**	[254,288]
			Cell line-derived xenografts	male SCID nude mice (6 wk), orthotopic; ***PDT regimen***: verteporfin, 690 nm, 74 mW/cm^2^, 10–40 J/cm^2^, 135–540 s	**Yes**	[245]
**PancTU-I**	PDAC	Unknown	Spheroids	nonadherent U-form plates with medium containing 20% methyl cellulose	No	[238]
			Cell line-derived xenografts	male or female SCID mice (13–20 wk), subcutaneous	No	[289]
			Cell line-derived xenografts	male or female SCID mice (13–20 wk), orthotopic	No	[289]
**PaTu 8902**	PAC	Primary	Spheroids	ultra-low attachment round-bottom plates	No	[252]
			Co-cultured spheroids	undifferentiated monocyte-like (THP-1) cells or THP-1 conditioned medium	No	[252]
			Cell line-derived xenografts	athymic nude mice, subcutaneous	No	[290]
**PaTu 8988**	PAC	Metastatic	Cell line-derived xenografts	male BALB/c nude mice (5–6 wk), subcutaneous	No	[291]
			Cell line-derived xenografts	BALB/c nude mice (5 wk), orthotopic	No	[291]
**PC-1**	PDAC	Metastatic	Cell line-derived xenografts	male or female NIH athymic nude mice (4–6 wk), subcutaneous	No	[292]
**PC-2**	PDAC	Metastatic	Spheroids	serum-free medium DMEM/F12 supplemented with bFGF, EGF, insulin, transferrin, sodium selenite, and bovine serum albumin	No	[293]
			Cell line-derived xenografts	male or female NIH athymic nude mice (4–6 wk), subcutaneous	No	[292]
**PC-3**	PDAC	Unknown	Cell line-derived xenografts	male BALB/c athymic nude mice (5 wk), subcutaneous	No	[294]
**PC-7**	PDAC	Unknown	Cell line-derived xenografts	female specific pathogen-free athymic nude mice (4 wk), subcutaneous	No	[295]
			Cell line-derived xenografts	BALB/c nude mice (5 wk), orthotopic	No	[291]
**PCI-24**	PAC	Primary	Cell line-derived xenografts	female BALB/c nude mice (4–6 wk), subcutaneous	No	[296]
**PCI-35**	PDAC	Primary	Cell line-derived xenografts	KSN Slc nude mice, subcutaneous	No	[297]
**PCI-43**	PAC	Primary	Cell line-derived xenografts	female BALB/c nude mice (4–6 wk), subcutaneous	No	[296]
**PDXPC1**	PDAC	Primary	Spheroids	serum-free medium DMEM/F12 supplemented with basic bFGF, EGF, and insulin	No	[298]
			Cell line-derived xenografts	female BALB/c nude mice (4–6 wk), subcutaneous	No	[298]
**PK-1**	PDAC	Metastatic	Cell line-derived xenografts	male BALB/c nude mice (5 wk), subcutaneous	No	[299]
**PK-45P**	PA	Unknown	Spheroids	ultra-low attachment round-bottom plates	No	[276]
**PK-8**	PDAC	Metastatic	Spheroids	ultra-low attachment plates	No	[300]
			Cell line-derived xenografts	SCID mice, subcutaneous	No	[301]
**PL45**	PAC	Primary	Spheroids	plates coated with 1% agarose in DMEM	No	[263]
			Cell line-derived xenografts	NOD/SCID nude mice (7–9 wk), subcutaneous	No	[302]
			Cell line-derived xenografts	athymic nude mice, orthotopic	No	[303]
**PSN1**	PAC	Primary	Spheroids	cancer stem cell medium	No	[254]
			Cell line-derived xenografts	male BALB/c nude mice (12–14 wk), subcutaneous	No	[304]
**PT45**	PDAC	Primary	Spheroids	gelatin porous microbeads	No	[305]
			Co-cultured spheroids	human normal fibroblasts or CAF	No	[305]
			Cell line-derived xenografts	C57BL athymic ICRF nude mice, subcutaneous	No	[306]
**PT45-P1**	PDAC	Primary	Spheroids	nonadherent round-bottom plates with medium containing 20% methyl cellulose	No	[238]
**S2-007**	PDAC	Metastatic	Spheroids	not listed	No	[307]
			Hydrogel-based spheroids	polypeptide network hydrogel	No	[308]
			Cell line-derived xenografts	BALB/c nude mice (8–16 wk), subcutaneous	No	[309]
			Cell line-derived xenografts	male athymic nude mice (5 wk), orthotopic	No	[307]
**S2-013**	PDAC	Metastatic	Co-cultured spheroids	HUVECs and human MSCs	No	[135]
			Organoids ^†^	HUVECs and MSCs	No	[135]
			Cell line-derived xenografts	female athymic BALB/cSlc-*nu*/*nu* mice (7 wk), subcutaneous	No	[135]
			Cell line-derived xenografts	female athymic nude mice (6–8 wk), orthotopic	No	[310]
			Organoid-based xenografts	S2-013 organoids, female athymic BALB/cSlc-*nu*/*nu* (7 wk), subcutaneous	No	[135]
**S2-020**	PDAC	Metastatic	Cell line-derived xenografts	BALB/c nude mice (8–16 wk), subcutaneous	No	[309]
**S2-028**	PDAC	Metastatic	Microfluidic spheroids	three-lane OrganoPlate channels	No	[311]
			Cell line-derived xenografts	BALB/c nude mice (8–16 wk), subcutaneous	No	[309]
			Cell line-derived xenografts	athymic mice, intrasplenic injection	No	[312]
**S2-CP8**	PDAC	Metastatic	Cell line-derived xenografts	male BALB/cAJcl nude mice (6 wk), orthotopic	No	[313]
**SK-PC-1**	PDAC	Unknown	Cell line-derived xenografts	female athymic BALB/c nude mice (5 wk), subcutaneous	No	[314]
**SU8686**	PAC	Metastatic	Cell line-derived xenografts	male BALB/cAJcl nude mice (6–8 wk), orthotopic	No	[315]
**Sui66-Sui70, Sui72-Sui74**	PDAC	Primary	Cell line-derived xenografts	female C.B.17/Icr Jcl-scid SCID mice (6–8 wk), subcutaneous	No	[316]
**Sui65, Sui71**	PDAC	Metastatic	Cell line-derived xenografts	female C.B.17/Icr Jcl-scid SCID mice (6–8 wk), subcutaneous	No	[316]
**SUIT-2**	PDAC	Metastatic	Co-cultured spheroids	PSCs	No	[284]
			Hydrogel-based spheroids	Matrigel and collagen I mixture (3:1)	No	[284]
			Cell line-derived xenografts	female BALB/c nude mice (6 wk), subcutaneous; ***PDT regimen***: rBC2-IR700, NIR light 670–710 nm, 100 J/cm^2^	**Yes**	[251]
			Cell line-derived xenografts	female nude mice (6 wk), co-implanted with PSCs, orthotopic	No	[317]
**SUIT-4**	PDAC	Metastatic	Cell line-derived xenografts	BALB/c athymic nude mice (6 wk), subcutaneous	No	[318]
**SUIT-58**	PDAC	Metastatic	Hydrogel-based spheroids	Collagen I and cell culture insert	No	[319]
**SW1990**	PAC	Primary	Spheroids	serum-free sphere medium DMEM/F12 supplemented with B27, bFGF, and EGF	No	[320]
			Cell line-derived xenografts	female BALB/c nude mice (5 wk), subcutaneous; ***PDT regimen***: quantum dots conjugated with integrin antagonist arginine-glycine-aspartic acid peptides, 630 nm, 100 mW/cm^2^, 1200 s	**Yes**	[321]
			Cell line-derived xenografts	male and female athymic N:NIH (S) nude mice (5–6 wk), orthotopic	No	[322]
**T3M-4**	PDAC	Primary	Spheroids	ultra-low attachment round-bottom plates	No	[276]
			Cell line-derived xenografts	BALB/c athymic nude mice (7 wk), subcutaneous	No	[323]
			Cell line-derived xenografts	female BALB/c athymic nude mice (6–8 wk), orthotopic	No	[324]
**TCC-Pan2**	PDAC	Metastatic	Cell line-derived xenografts	female BALB/c nude mice (4 wk), orthotopic	No	[282]
**YAPC**	PA	Metastatic	Spheroids	hanging drop method	No	[325]
			Cell line-derived xenografts	male NMRI mice (4–6 wk), subcutaneous	No	[326]

Abbreviations (alphabetical): bFGF, basic fibroblast growth factor; BPD, benzoporphyrin derivative; BME, basement membrane extract; CAF, cancer-associated fibroblasts; Ce6, chlorin e6; DMEM, Dulbecco’s modified Eagle’s medium; EGF, epidermal growth factor; GRHL2, grainyhead like transcription factor 2; HUVECs, human umbilical vein endothelial cells; MSCs, mesenchymal stem cells; NIR, near-infrared; PA, pancreatic carcinoma; PAC, pancreatic adenocarcinoma; PASC: pancreatic adenosquamous carcinoma; PDAC, pancreatic ductal adenocarcinoma; PDT, photodynamic therapy; PEG, polyethylene glycol; PSCs, pancreatic stellate cells; wk, weeks. ^†^ Organoid classification ambiguous because the models did not conform to the full definition of organoids. Cell information was retrieved from DepMap and Cellosaurus.

### 3.4. Challenges and Caveats of Biomimetic PDAC Models in the Context of PDT

A generally applicable rule of thumb in preclinical research is that the more remote a model is from the actual human condition, the less representative the model outcomes will be relative to that human condition [327]. It is therefore important to appreciate that models will never provide full representation of how in situ tumours behave in patients with respect to locoregional biology, biochemistry, physiology, and pharmacodynamic responsiveness. On the spectrum of available models covered in this paper, using cell monolayers (2-D cultures) is associated with the highest probability of achieving results with least translational potential, whereas working with PDXs in a quasi-native milieu (e.g., transplanted into an animal, preferably orthotopically) will probabilistically furnish the highest level of representation. This phenomenon was recently illustrated by Gioeli et al. [328], where RNAseq and subsequent protein–protein interaction networks analysis of cells in 2-D cultures and PDX were compared against patients’ tumour biopsies. With tumour biopsies as the transcriptomic baseline, the analyses revealed that 2-D cultures (Figure 8A) exhibited the highest number of differentially expressed transcripts versus PDX (Figure 8B) and 3-D cell co-culture (Figure 8C). These protein–protein interaction networks featured processes that are relevant to PDAC (patho)biology, including oxidative stress, immune function, cell cycle, and ECM interaction. Accordingly, and with respect to biomimetic models, maximal emulation of numerous facets of human tumours in their natural environment is warranted, which necessitates the veering away from simple systems when robust data are a priority.

The complexity involved in the creation of a near-native biomimetic in vitro PDAC model is exemplified in the work by Gioeli et al. [328]. The authors used a dual compartment, porous polycarbonate transwell membrane culture system where human primary PSCs and PDAC-derived cancer cells were plated in spheroid format on the bottom end of the membrane and primary human microvascular endothelial cells were plated in monolayer format on the upper end of the membrane. The top surface of the membrane was coated with gelatin and the bottom surface with collagen, and cells were plated in sequential order and cultured under artificial flow conditions to emulate hemodynamics [331] in the ‘vascular’ compartment and interstitial flow dynamics [332,333] in the ‘parenchymal’ compartment. This culture system, which is more complex than the biomimetic models described above, not only resembles PDXs on a transcriptional level [328], but most accurately mimics the in situ tumour transcriptome (Figure 8C) surprisingly more so than PDXs (Figure 8B). The system is suitable to test drug delivery, pharmacodyamics, toxicity, and resistance mechanisms. A system of this calibre is not available to most researchers, which is a caveat in and of itself. Another drawback of in vitro models is that human PDAC is characterised by hyperdense stroma that is hypovascular in the juxta-tumoural and pan-stromal areas, i.e., tissues directly adjacent to the tumour [334], which is a feature that is commonly not emulated in PDAC models and mostly affects drug delivery parameters. Normally, systemically administered (photo)drugs and nanomedicines have to extravasate into the hypervascular normal adjacent pancreatic tissue and transverse a relatively large distance to reach PDAC cells. These drug migration obstacles are commonly discounted in the in vitro and ex vivo models described in this review.

On top of the challenge of technical availability and accessibility, in vitro and in vivo biomimetic models are associated with additional caveats when employed for PDT research. Both models (in cases where human PDAC cell lines are used) lack a complete immune system that is instrumental in ECM shaping, immunotolerance, cell–cell interactions, and anti-tumour immune responses after PDT [335,336,337]. This not only has a bearing on PS/drug delivery to the tumour cells, which is expected to be easier in especially the in vitro models due to the absence of phagocytic cells, but also skews therapeutic responses. Subsets of innate immune cells such as neutrophils, monocytes, and macrophages typically constitute the first wave of resident and migratory cells to clean up oxidatively afflicted cells and debris [338,339,340,341], whereas subsets of the adaptive immune system (e.g., dendritic cells and T cells) are required for long-term tumour control [338,342,343,344,345]. Post-PDT immune signals in the form of chemokines, cytokines, and other classes of immunomodulatory mediators [346] emitted by resident and chemoattracted immune cells [338] as well as PDT-treated tumour cells [347] therefore do not materialise at all in in vitro and ex vivo models that completely lack immunological constituents, or materialise partially yet insufficiently in nude mice [342,343,344] that lack T lymphocytes but do produce innate immune cells [348]. Effectuation of immune system signalling networks and PDT-pertinent immunobiology could be achieved in syngeneic tumour models employed in combination with orthotopic cell transplantation (Table 3), granted that thus obtained tumours sustain a high level of biomimetic integrity relative to their in situ human counterparts.

Another caveat in the PDAC models when used in PDT research is atmospheric composition, which should change to mirror post-PDT conditions, and the ramifications on PDAC (patho)biology. The in vitro and ex vivo models typically do not account for the switch from normoxia to conditions of low partial pressure of oxygen (pO_2_) after PDT. PDT causes a conversion of molecular oxygen to superoxide anion or singlet oxygen [360] (Figure 7), which coincides with a rapid reduction in pO_2_ and reduced tumour oxygenation [361]. On top of that, a fraction of the PS molecules is taken up by endothelial cells following injection. Upon PDT, photosensitised endothelial cells are damaged and become thrombogenic [362,363,364] owing to photochemical disruption of cell integrity, endothelial cell activation, and exposure of the basement membrane. The corollary vascular shutdown in turn produces prolonged hypoxia or anoxia in tumour cells, metabolic catastrophe [169], and cell death as well as activation of cell survival pathways [158,197]. Moreover, PDT-treated tumours become more acidic as a result of the hypoxia as lactic acid production and build-up increase due to lacking substrate for aerobic respiration (i.e., O_2_). As explained in Section 2, acidic conditions select PDAC cells with a more aggressive phenotype geared towards invasion and metastasis [37]. In contrast, although intracellular pO_2_ drops during PDT in in vitro and ex vivo models, the cells and tissues are normally maintained in an atmosphere composed of 95% air (containing 20% molecular oxygen) and 5% carbon dioxide (incubator) or 100% air (outside of an incubator), accounting for virtually immediate reoxygenation upon cessation of light exposure. Consequently, the above-referenced features should be incorporated into PDT studies using PDAC models to properly contextualise the research and improve the biomimetic character of the models. This can be achieved by using specialised culture chambers with tunable atmospheric conditions [182,195,196].

The more advanced models where PDACs are grown volumetrically, such as spheroids and organoids, may suffer from drug distribution heterogeneity. Many anti-cancer drugs, including first- and second-generation PSs [167], are lipophilic and hence preferentially localise to cell and organelle membranes [170,365]. Photosensitisation of these PDAC cell clusters occurs by adding the PS to the culture medium and allowing the PS molecules to disperse throughout the cell cluster. However, as shown for spheroids in Figure 9, lipophilic PSs tend to have a predominantly peripheral accumulation pattern in which the core of the cell cluster becomes less photosensitised. Inasmuch as ROS are generated only at loci where PS molecules and oxygen coincide, the core of the spheroid would exhibit less therapeutic effect following PDT. In terms of distribution, the same applies to nanoformulated (liposome-encapsulated) lipophilic PSs (Figure 9) as well as amphipathic compounds such as Hoechst 33342 (Figure 9) but not highly hydrophilic PS derivatives, which tend to penetrate spheroids homogenously (unpublished results). One could argue that a similar distribution pattern will manifest in vivo, where PS molecules or their delivery vehicles are fed to the PDAC through remote extravasation points as explained above.

Finally, tumours grown in vivo are exposed to a partial or full arsenal of immune cells following initial damage that will remove dead and dying cells from the tumour volume and, in case of syngeneic models, facilitate abscopal immunological cell death in residually viable remnants of the PDAC [210,366,367]. These key phenomena are not captured in the in vitro and ex vivo models described here, which could skew experiment results obtained with biomimetic PDAC models.

### 3.5. Future Directions

The challenges described in the previous section regarding biomimetic pancreatic cancer models for a large part shape the future directions. Accordingly, the usually powerful KISS principle (keep it simple, stupid!) does not apply to biomimetic PDAC models. Instead, increasing model complexity is warranted, especially in regard to improving the models’ physiologic and biochemical accuracy and relevance to human disease. The first key direction is the formation of more complex TMEs with precise control over cell types and matrix composition, which can be achieved by e.g., 3-D bioprinting [368,369,370]. Secondly, this approach must be combined with steps to better emulate the interactions between cancer cells, stromal cells, and immunological constituents, which can be realised through the use of co-culture systems. It should be noted that the first two points only apply to spheroids and organoids since PDXs and other forms of ex vivo models with tumour tissue do not allow for such engineering. Thirdly, the cellular components that make up the PDAC and TME require adequate provision of nutrients and exposure to mechanical stress that can be provided by tumour-on-a-chip models [371,372,373] or bespoke 3-D spheroid systems that incorporate the abovementioned factors (Figure 8) [328] as well as those that can control atmospheric composition [374,375], given that hypoxic conditions are instrumental in both tumour biology [376] and post-PDT responses [158,377]. Finally, PDACs are generally heterogeneous at the genetic-, cellular-, and histological level [378,379], not only intertumourally but also intratumourally [378], which significantly augments treatment difficulty [380]. This aspect has also been reported for other types of hepatopancreaticobiliary tumours such as cholangiocarcinoma [381], equally accounting for its tenacious therapeutic recalcitrance [382,383]. Consequently, tailoring models based on patient-specific tumour data for more effective drug screening and individualised therapies cannot be dismissed and is of particular importance in oncological settings that are centred on personalised medicine.

As to future directions of PDT in the context of biomimetic PDAC models, the following points are noteworthy. Most importantly, several variables, such as circulation, systemic chemotactic gradients, and recruitment of innate and adaptive immune cells to the PDT-treated tumour are absent in all models except for syngeneic tumour models in immunocompetent animals. As alluded to previously, these variables play a quintessential role in therapeutic efficacy and long-term tumour control. In cases where therapeutic efficacy assessment constitutes the main objective of a study, it is recommended that models are used that encapsulate these variables to generate maximally representative results. Guidance on such models is provided in Table 3. Secondly, PDT data produced using in vitro and ex vivo models should ideally be validated in proper animal models of cancer. As an example, complete PS distribution across the tumour volume is critical inasmuch as therapeutic and (patho)biological responses are reliant on the degree of intratumoural photosensitisation [158,384]. Lack of sufficient mimicry could cause a divergence in outcomes and conclusions between in vitro/ex vivo models and in vivo models that are characterised by tumour eradication (complete tumour photosensitisation) in the former versus tumour survival (inadequate or heterogeneous photosensitisation) in the latter. Naturally, this potential issue does not apply to 2-D PDAC cultures, which in our opinion harness little translational utility and should only be employed for the most rudimentary tests. For 3-D tumours it is possible, even likely, that the waning degree of photosensitisation with depth (e.g., Figure 9) also prevails in PDXs, ex vivo models, and in vivo (xenografts). Tumour photosensitisation in vivo is dependent on a plethora of factors, including intratumoural vascularisation, PS or nanoparticulate PS pharmacokinetics and disposition, interstitial fluid flow, and desmoplasia. The first three factors are not recreated in in vitro and ex vivo PDAC models per se, which may lead to an overestimation of the degree of photosensitisation in these models and bias the data in favour of tumour eradication. Furthermore, intratumoural distribution varies with the type of PS or PS delivery system employed. A particular PS therefore does not serve as a template for other PSs, with nano-encapsulation in and of itself being able to fundamentally alter pharmacokinetics, pharmacodynamics, biodistribution, intracellular localisation, and disposition of the PS [385,386,387]. Also, mapping PS distribution profiles in models should preferably be verified using orthotopic tumour xenografts (Table 2 and Table 3). Subcutaneous xenografts may differ from tumours cultivated in their native environment and conditions in terms of, e.g., microvascularisation and angiogenesis [388,389,390], which comprise the conduit networks that facilitate PS delivery into tumours and hence partially dictate photosensitisation.

Lastly, future directions should be geared towards optimisation of technical aspects of test models. PDT is a modality devised to eradicate cancer cells; measuring cell viability is therefore the primary outcome parameter in many studies whilst functional molecular probes are also used for other purposes such as qualifying and/or quantifying the events that culminate in tumour cell death. As discussed above and shown in Figure 9, the penetration of amphipathic and lipophilic molecules is hampered in 3-D models, which includes reporter dyes such as Hoechst 33342 (nuclear stain) and likely extends to other classes of commonly used molecular reporters such as cell viability probes (e.g., MTT [391], PubChem CID 16218671, logP = 5.9), organelle-specific probes for localisation studies (e.g., MitoTracker Red [196], PubChem CID 22613925, logP = 1.2, fluorescent probe for mitochondria; LysoTracker Red [392], PubChem CID 15410449, logP = 2.1, fluorescent probe for lysosomes [393]), cell and organelle membrane stains (e.g., DiOC6(3) [394], PubChem CID 9894321, logP = 5.5–6.5), mitochondrial membrane depolarisation probes (e.g., JC-1 [395], PubChem CID 5492929, logP range = 3–5), and fluorogenic redox probes to measure ROS (e.g., DCFH_2_(-DA) [162,229,396], PubChem CID 77718, logP = 4.4–4.6). As a result of the incomplete penetration of molecular probes, the readouts may underestimate effect size or paint a partial picture. These factors should be accounted for in experimental design and measures should be taken to remediate technical hurdles, such as using water-soluble alternatives (e.g., WST-1 instead of MTT to determine cell viability), and protocols should ideally be standardised for the complex models. Secondly, PDT with certain PSs is associated with vascular shutdown [249,397,398] that leads to intratumoural hypoxia/anoxia, which activates survival pathways that in turn affect therapeutic outcome [158,169,182,197]. Accordingly, these treatment-induced pathophysiological conditions should be accounted for in experimental design by equipping the test systems with atmospheric control [195,196,377]. Thirdly, factors that are not representative of the clinical situation yet deleterious to PDT outcomes should be avoided in experimental designs. For example, we have found that mycoplasma infection substantially distorts the molecular biological responses to PDT in tumour cells as well as the temperature at which PDT is performed, which should be body temperature instead of the frequently used room temperature. We are preparing manuscripts that address both issues and provide guidelines for proper experimental design.

## 4. Conclusions

Although the use of 2-D cell lines has been instrumental for basic in vitro PDAC research, improved 3-D models offer several inherent advantages that move the models towards an in situ biomimetic character. Primarily, the PDAC TME can be represented more faithfully, which facilitates a more accurate investigation of PDAC biology and novel treatment strategies that affect the TME. Ultimately, using representative models under clinically representative conditions is expected to increase the rate of clinical translation and more effectively address the need to improve PDAC treatment outcomes. Evolving technologies may push the representability of biomimetic PDAC models closer to in situ tumours in humans, while standardisation of the models will allow for intra- and interstudy comparisons.

These predicates are particularly relevant for PDT, an emerging minimally invasive treatment modality that kills photosensitised tumour cells directly through hyperoxidative stress and indirectly through treatment-induced immunological cell death. PDT has the capacity to destroy the TME–a feature of PDAC that has been attributed to therapeutic recalcitrance–and resolve non-treated tumours (e.g., metastases of the same phenotype as the treated tumours) via abscopal effects. Consequently, PDT is a potentially useful treatment strategy for PDAC.

Unfortunately, the majority of PDAC research that focuses on PDT has been performed in 2-D models, which we have deemed to have the lowest level of translational value. In this review we therefore addressed more appropriate biomimetic research models for PDT, including xenografts, PDX, and spheroids and elaborated on the advantages and disadvantages of these models. The most important challenges and caveats of biomimetic PDAC models include the necessity to reconstruct a pleiotropic TME that accounts for photosensitisation and probe gradients as well as differential pO_2_ levels across the 3-D models. PDT leads to hypoxia, which in itself is associated with a particular gamut of signalling cascades that add to the post-therapeutic molecular hyperoxidative stress-affected landscape and have the capacity to alter treatment outcomes. In case of xenografts, orthotopic syngeneic PDAC models are favourable due to the presence of a native pancreas milieu and functional immune system, which is mandatory for long-term tumour control. Finally, the finetuning of methodological approaches and standardization of experimental protocols is warranted to ensure interstudy comparisons.

## Figures and Tables

**Figure 1 ijms-26-06388-f001:**
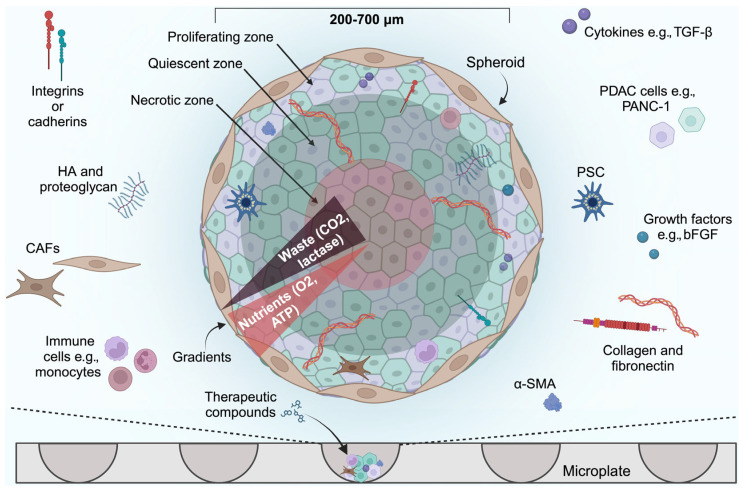
The pancreatic ductal adenocarcinoma (PDAC) stromal microenvironment can be replicated in vitro using 3-D spheroid structures. Spheroids can be formed using a low-adhesion microplate and consist of proliferating PDAC cells, quiescent cells, and a hypoxic core that is acidic due to lactate build-up. When co-cultured, additional cell types are introduced to better replicate the PDAC tumour microenvironment (TME). For example, cancer-associated fibroblasts (CAFs) can help to model the dense stroma by producing several of its constituents, such as the structural protein collagen, though such components can also be incorporated experimentally. Immune cells can also be used to help model the immunosuppressive features observed in PDAC. PDAC cells themselves express many of the extracellular matrix (ECM) proteins, including integrins and cadherins, which span the membrane and mediate adhesion to other PDAC cells and ECM components. Abbreviations: 3-D, 3-dimensional; α-SMA, alpha smooth muscle actin; ATP, adenosine triphosphate; bFGF, basic fibroblast growth factor; HA, hyaluronic acid; PSC, pancreatic stellate cell; TGF-β, transforming growth factor beta. Figure was made on Biorender.com.

**Figure 2 ijms-26-06388-f002:**
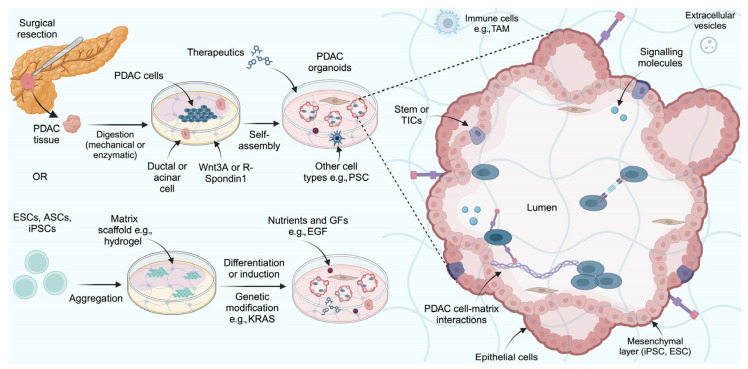
The structure and function of pancreatic ductal adenocarcinoma (PDAC) organoids. PDAC organoids are commonly derived from resected tumour samples or stem cells with the potential to form a structure that is able to recapitulate specific morphological and structural features of the malignancy. As organoids can be engineered to contain various PDAC tumour microenvironment (TME) components including fibroblasts, matrix proteins, and immune cells, they are able to replicate endogenous TME cell functions and heterogeneity. The optional use of naturally derived or synthetic polymers in culture can recreate the 3-D extracellular matrix (ECM) composition as the PDAC and epithelial (ductal or acinar) cells comprising the organoid interact with these components via receptor proteins such as integrins. Within the organoid, a large population of PDAC cells send and receive signals from the matrix and orchestrate both paracrine and autocrine signalling using a variety of signalling molecules, including the cytokine transforming growth factor beta (TGF-β). Abbreviations: 3-D, 3-dimensional; ASC, adult stem cell; EGF, epidermal growth factor; ESC, embryonic stem cell; GF, growth factor; iPSC, induced pluripotent stem cell; KRAS, Ki-ras2 Kirsten rat sarcoma viral oncogene homolog; PSC, pancreatic stellate cell; TAM, tumour-associated macrophage; TIC, tumour initiating cell. Figure was made on Biorender.com.

**Figure 3 ijms-26-06388-f003:**
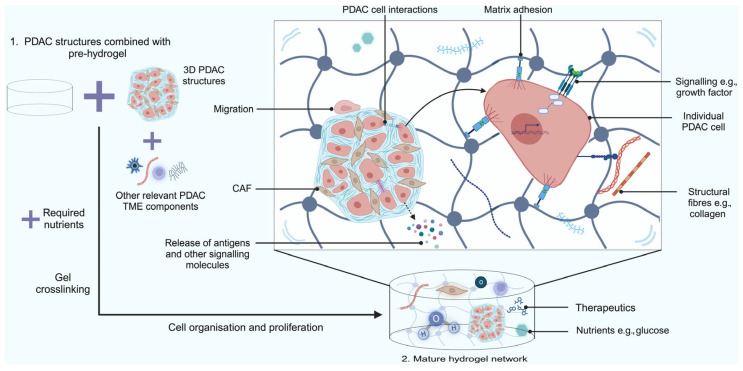
Three-dimensional hydrogel scaffold models of pancreatic ductal adenocarcinoma (PDAC). Various PDAC structures, including spheroids or organoids, can be inserted into hydrogel scaffolding, which comprises covalently bound hydrophilic polymers arranged in a 3-D network with water, oxygen, and nutrients. Both within and between the 3-D structures, PDAC cells biochemically and physically communicate via paracrine (signal secretion) or physical (junctions between neighbouring cells) mechanisms. Signals are received and internalised, which leads to gene transcription changes that reflect those observed in vivo. Similarly, PDAC cells interact with the scaffolding via adhesion proteins such as integrins as well as actin, which facilitates the conversion of mechanical forces from the matrix into biochemical signals that enter the PDAC cell nucleus. In many models, the stiffness of the matrix is tunable and facilitates the proliferation and migration of PDAC cells via focal adhesions to extracellular matrix (ECM) proteins. The matrix also enables the diffusion of soluble factors and signalling molecules. Abbreviations: CAF, cancer associated fibroblast; GF, growth factor; HA, hyaluronic acid; PSC, pancreatic stellate cell; TGF-β, transforming growth factor beta. Figure was made on Biorender.com.

**Figure 4 ijms-26-06388-f004:**
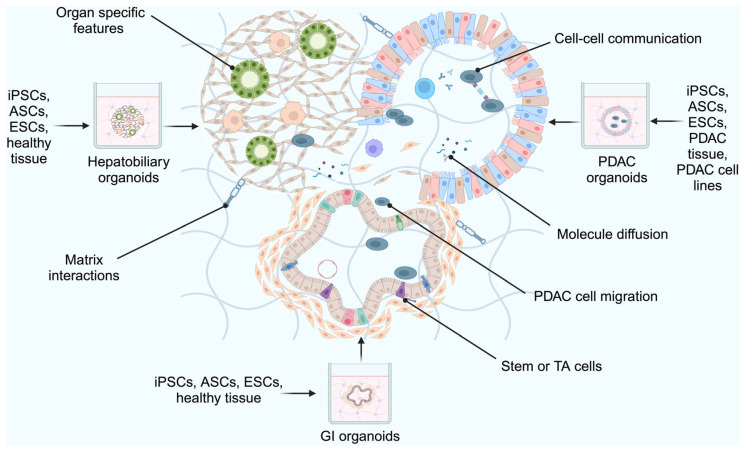
Exemplary 3-D assembloid model use for in vitro studies of pancreatic ductal adenocarcinoma (PDAC). Assembloid structures are most commonly formed by fusing multiple organoids that represent different organ systems. When cultured, additional relevant cell types can be included such as endothelial cells, which help form immature vasculature, and immune cells, which help replicate the immune tumour microenvironment in PDAC. These multi-region assembloids are unique in that they display a high level of connectivity and can demonstrate the interaction between organ-specific cell types. In particular, the movement of PDAC cells in between organoid types represents the process of metastasis to nearby organs. The diffusion of drugs, nutrients, oxygen, and metabolic waste can be modelled, as well as the interaction of the organoids with the incorporated extracellular matrix (ECM) components. Abbreviations: ASC, adult stem cell; ESC, embryonic stem cell; GI, gastrointestinal; iPSC, induced pluripotent stem cell; TA, transit amplifying. Figure was made on Biorender.com.

**Figure 5 ijms-26-06388-f005:**
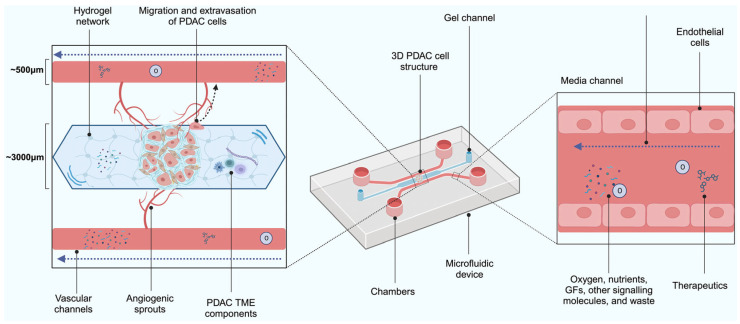
Microfluidic pancreatic ductal adenocarcinoma (PDAC) models. Three-dimensional (3-D) in vitro PDAC structures, including spheroids and organoids, can be cultured using microfluidic devices that contain miniature circuits with precise dimension. The circuit mimics the vasculature observed in the PDAC tumour microenvironment (TME), with the ‘vessels’ being lined with endothelial cells. Angiogenic sprouts can carry molecules to and from the PDAC cells and aid in the creation of gradients similar to those observed in vivo. The use of a co-cultured spheroid, which can include other cell types seen in the PDAC TME such as pancreatic stellate cells (PSCs), can help model the environment with even greater accuracy. Pharmaceutics can be injected into the microcircuit. One important use of such a technique is to investigate anti-tumour drug delivery through the PDAC stroma. In addition, the small scale of the device enables controlled drug flow to the PDAC structures, which facilitates the assessment of drug concentrations (e.g., efficacy and cytotoxicity) more precisely. The dotted arrows represent the flow direction of media. Abbreviations: GF, growth factor. Figure was made on Biorender.com.

**Figure 6 ijms-26-06388-f006:**
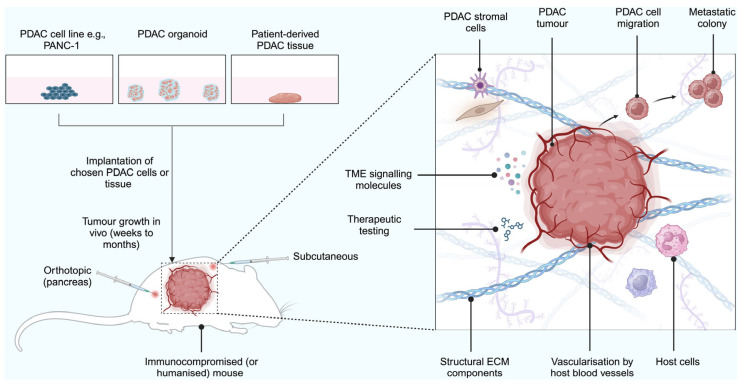
Pancreatic ductal adenocarcinoma (PDAC) xenograft models. Xenotransplantation involves the transfer of PDAC cell lines, PDAC organoids, or patient tissue into a recipient, usually an immunocompromised mouse. Though the establishment and growth of the cells in vivo is a long process (typically weeks), the developed tumour interacts with the graft and host tumour microenvironment (TME) components and displays a high similarity to PDAC disease progression in humans. The model is considered high fidelity owing to the vascularisation of the tumour by the host’s blood vessels, which facilitate the delivery of various molecules, including any therapeutics, to the tumour. As well as the spatial structure of PDAC being represented, use of patient tissue can retain intratumour heterogeneity and tumour subtype. Abbreviations: ECM, extracellular matrix. Figure was made on Biorender.com.

**Figure 7 ijms-26-06388-f007:**
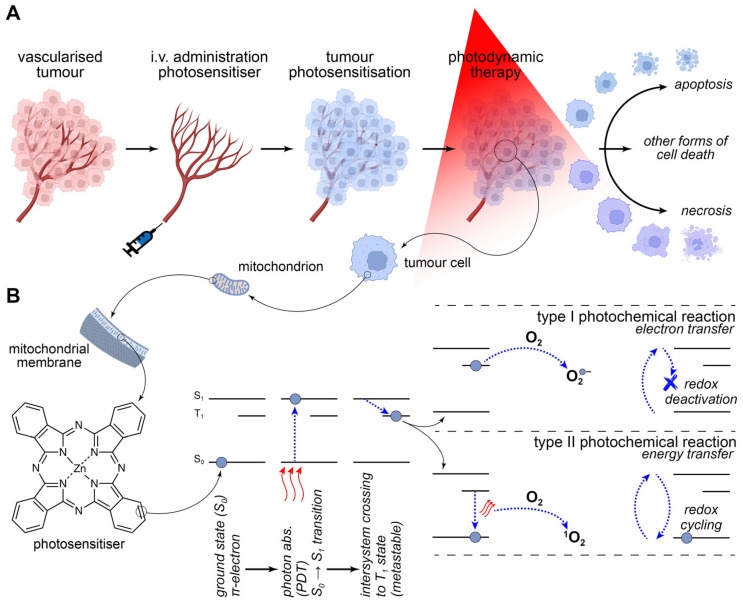
Mechanistic overview of photodynamic therapy (PDT). (**A**) PDT is used for the treatment of vascularised tumours inasmuch as intratumoural vascularisation is required to deliver intravenously administered photosensitiser (PS) molecules into the tumour and to ensure locoregional oxygenation. Tumour photosensitisation occurs once PS molecules have accumulated in the tumour, upon which the tumour is illuminated with light (i.e., PDT, typically at a wavelength of ≥630 nm) to activate the photosensitiser (**B**). This process culminates in mainly apoptosis and necrosis, but also other forms of cell death, that are chiefly responsible for post-therapeutic tumour removal. (**B**) Sequential zooming in from a tumour cell ((**A**), encircled) → mitochondrion → mitochondrial membrane → membrane-embedded PS → π-electron leads up to the visualisation of quantum chemical events during PDT. In the presented sequence of Jablonski diagrams, a ground state (S_0_) electron is elevated to the first excited state (S_1_) following the absorption of a photon (wavy red arrows) at (near-)resonant frequency. The excited state electron undergoes intersystem crossing from the S_1_ state to a triplet state (T_1_), from which one of two possible photochemical reactions takes place in the presence of a substrate—typically molecular oxygen (O_2_). In case of type I photochemical reactions, the triplet state electron is transferred to O_2_ to form superoxide anion (O_2_^•−^) as primary reactive oxygen species (ROS). Since the PS loses an electron, the molecular structure of the PS changes and with it the spectral properties, which often results in an ipsochromic shift of the absorption band and a reduction in triplet state quantum yield at unaltered illumination wavelength (designated as redox deactivation). Alternatively, type II photochemical reactions are characterised by a T_1_ → S_0_ transition of the electron with concurrent energy transfer to O_2_. This leads to the formation of singlet oxygen (^1^O_2_), a highly cytotoxic ROS, with preservation of the redox cycle, meaning that the electron can repeat the photo-excitation and decay process ad infinitum as long as photons are absorbed and O_2_ is present in the direct vicinity of the PS.

**Figure 8 ijms-26-06388-f008:**
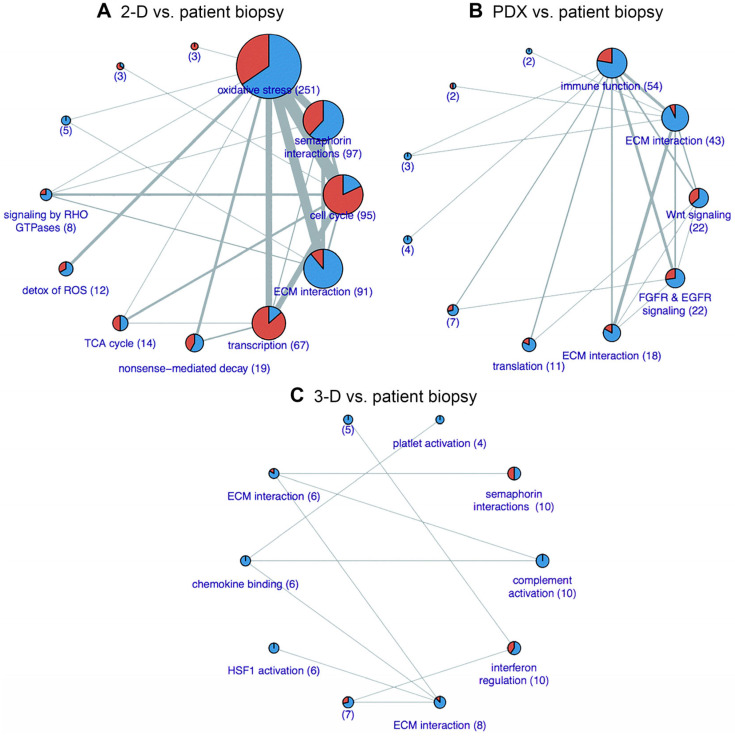
Molecular representability of biomimetic pancreatic ductal adenocarcinoma (PDAC) models. PDAC tissue biopsies were obtained perioperatively from a surgical patient and subjected to RNA sequencing (RNAseq). RNAseq was also performed on cell culture monolayers [329], orthotopic patient-derived xenografts (PDXs) in immunocompromised (athymic) mice [330], and 3-D cell co-cultures [328], all created with cells from the same tumour biopsy (referred to as PDAC cell line 449). The technical aspects of the 3-D culture are explained in the text of this section. Differential expression of genes was analysed and expressed against the patient biopsy transcriptome. Data were plugged into protein–protein interaction networks derived from the STRING database and further processed by partitioning into communities (set of proteins in nodes (spheres) with upregulated (red) and downregulated (blue) genes). The larger communities are specified and the size of each node is scaled to the number of transcriptionally dysregulated proteins in that community. Communities are connected by gray lines to reflect their relationship, while line thickness indicates the degree to which the internode relationship is affected at the transcriptional level. Presented are the differential expression profiles of 2-D cell culture: ((**A**), 665 differentially regulated proteins), PDX; ((**B**), 188 differentially regulated proteins), and the 3-D cell co-culture ((**C**), 72 differentially regulated proteins) relative to the respective clinical biopsy transcript levels. This figure was reproduced from [328] in accordance with creative commons attribution-noncommercial 3.0 unported licence and following written permission from the corresponding author.

**Figure 9 ijms-26-06388-f009:**
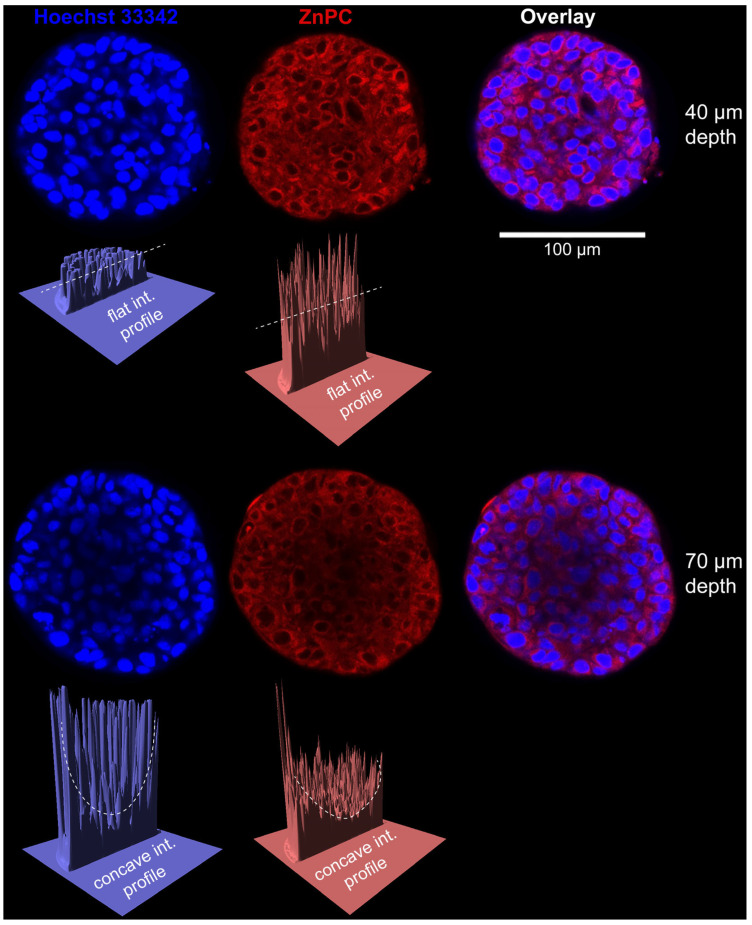
Heterogeneous distribution pattern of Hoechst 33342 and liposomal zinc phthalocyanine (ZnPC) in human extrahepatic cholangiocarcinoma (TFK-1) spheroids. Confocal microscopy-generated planar fluorescence profiles are shown that were acquired at a depth of 40 µm (top row) and 70 µm (bottom row) from the top surface of the spheroid. The fluorescence intensity patterns demonstrate relatively equal distribution of Hoechst 33342 (blue) and liposomal ZnPC (red) across the spheroid in the top plane and tapered penetration towards the spheroid core in the deeper situated plane, as indicated by the dashed lines. The images were quantitatively processed with custom-built image analysis software (CanAlysis version 1, Applive, Amsterdam, The Netherlands), where pixel intensity maps were created for the blue and red hues. Spheroid planes were cross-sectioned at the longest diameter and one half of the plane was tilted to reveal the red or blue intensity landscape (*z*-axis; 0–255-pixel intensity range) across the spheroid plane (x- and y-axes). Whereas the tissue plane at 40 µm depth had a rather flat intensity profile, the intensity profile of the 70 µm tissue plane was concave. The latter indicates lower fluorescence intensities in the centre of the spheroid (i.e., farthest diffusion distance from the outer spheroid surface) and hence limited penetration of lipophilic and amphipathic molecules into the spheroid. TFK-1 spheroids were prepared by liquid overlay in a Nunc Lab-Tek II Chamber Slide System (8-well format). A thin layer of 100% Matrigel was added to each well and incubated at 37 °C for 30 min under standard culture conditions. Subsequently, 4 × 10^5^ TFK-1 cells in 2% Matrigel were added and incubated for 4 d. Medium was refreshed every 2 d. On day 4, spheroids were incubated with ZnPC-loaded PEGylated lecithin liposomes (0.003 ZnPC:phospholipid molar ratio [170,365], 5 µM final ZnPC concentration) for 24 h. Then, spheroids were washed with PBS and fixed with 2% paraformaldehyde/1% glutaraldehyde in PBS for 30 min. Spheroids were washed 3 × with PBS, followed by quenching of background fluorescence with 0.1% NaBH_4_ for 10 min. Nuclei were stained with Hoechst 33342 (32 µM final concentration; log P = 4.6, PubChem CID 1464) for 2 h and slides were mounted with SlowFade Gold. Imaging was performed with a Leica TCS SP8 system using a 25 × water immersion objective. ZnPC (log P = 8.5 [167]) was imaged owing to its weak autofluorescence [163].

**Table 1 ijms-26-06388-t001:** Summary of PDAC models and their characteristics.

Model	Spheroid	Co-Cultured Spheroids	Microfluidic Spheroid Models	Hydrogel Scaffold-Based Models	Organoids	Assembloids	Organoid-Based Xenografts	Patient-Derived Xenografts	Cell Line-Derived Xenografts
**Application**	Investigating PDAC drug toxicity	Investigating PDAC drug toxicity	Industrial, large-scale PDAC drug testing	Testing PDAC drugs where loss of cell viability is not an issue	Investigating the physiology of PDAC	Investigating physiological changes in multiple organs	Investigating disease progression in PDAC	Investigating the efficacy of PDAC therapeutics (TME is well represented)	More convenient than PDXs, but do not fully represent the TME
**Cost**	Low	Moderate	Low	Moderate-high	Moderate	Moderate-high	Moderate-high	High	Moderate-high
**Time**	~3 to 5 days	~3 to 5 days	Within 7 days	~10 days	~4 days	≥7 days	1–2 months	Up to 6 months	1–2 months
**Specimen**	Cell lines	Cell lines	Cell lines	Cell lines	Tumour tissue, tumour cells, or stem cells	Tumour tissue, tumour cells, or stem cells	Tumour tissue, tumour cells, or stem cells and immunocompromised mice	Resected tumour tissue and immunocompromised mice	Cell lines and immunocompromised mice

**Table 3 ijms-26-06388-t003:** Non-exhaustive summary of non-human PDAC cell line-based models used in PDT research.

Cell Line	Disease	Source	Models	Methods	Tested in PDT	Ref.
Mouse						
**6606PDA**	PDAC	Primary	Cell line-derived xenografts	male C57BL/6 mice (6–8 wk), orthotopic	No	[349]
**K8484**	PDAC	Primary	Cell line-derived xenografts	KPC mice, subcutaneous	No	[350]
**KPC3**	PN	Primary	Cell line-derived xenografts	male C57BL/6 mice (8–10 wk), subcutaneous	No	[351]
**Panc02**	PDAC	Primary	Co-culture spheroids	Matrigel, co-culture with CD8^+^ cytotoxic T cells	No	[352]
			Hydrogel-based spheroids	Matrigel and complete DMEM mixture; Matrigel and collagen type I mixture	No	[352,353]
			Cell line-derived xenografts	female C57BL/6 mice (6–8 wk), subcutaneous; C57BL/6 mice (6–8 wk), orthotopic; ***PDT regimen***: IR700-conjugated anti-CD44 monoclonal antibody, 690 nm, 150 mW/cm^2^, 50 J/cm^2^; PTT regimen: 980 nm, 850 mW/cm^2^, 50 J/cm^2^, 600 s	**Yes**	[354,355]
			Cell line-derived xenografts	C57BL/6 mice (6–8 wk), orthotopic; PTT regimen: 980 nm, 850 mW/cm^2^, 50 J/cm^2^, 600 s	PTT	[355]
**UN-KC-6141**	PDAC	Primary	Cell line-derived xenografts	C57BL/6 mice, orthotopic	No	[356]
**UN-KPC-960**	PDAC	Primary	Cell line-derived xenografts	B6.129 mice, orthotopic	No	[356]
**UN-KPC-961**	PDAC	Primary	Cell line-derived xenografts	B6.129 mice, subcutaneous and orthotopic	No	[356]
**Hamster**						
**HaP-T1**	PDAC	Primary	Cell line-derived xenografts	male Syrian golden hamsters (5 wk), orthotopic	No	[322]
**PC-1.0**	PDAC	Primary	Cell line-derived xenografts	male Syrian golden hamsters (5 wk), orthotopic	No	[357]
**PC-1.2**	PDAC	Primary	Cell line-derived xenografts	Syrian golden hamsters (8 wk), orthotopic	No	[358]
**WD PaCa**	PDAC	Primary			No	[359]

Abbreviations (alphabetical): DMEM, Dulbecco’s modified Eagle’s medium; PDAC, pancreatic ductal adenocarcinoma; PDT, photodynamic therapy; PTT, photothermal therapy; wk, weeks.

## Data Availability

No new data eligible for sharing were created or analysed in this study. Data sharing is not applicable to this article.

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
