# Peer review of "Biomimetic Tumour Model Systems for Pancreatic Ductal Adenocarcinoma in Relation to Photodynamic Therapy"

_ijms, 2025, doi:10.3390/ijms26136388_

Round 1

Reviewer 1 Report

Comments and Suggestions for Authors

The manuscript by Olivia M. Smith et al. “Biomimetic Tumour Model Systems for Pancreatic Ductal Adenocarcinoma in Relation to Photodynamic Therapy” focuses on the existing preclinical models for human PDAC and discuss advancements in tissue remodelling to guide translational PDAC research. Although the review is well-written, there are some limitations as mentioned below: 

  1. The authors should deeply clarify the mechanism of photodynamic therapy.
  2. The authors should add more information in “Conclusions”.
  3. The authors should add the full names in Figure legends.

Reviewer 2 Report

Comments and Suggestions for Authors

Brief summary

In this review, Smith and colleagues summarize existing preclinical models for human pancreatic ductal adenocarcinoma (PDAC) and discuss advances in tissue remodeling to enhance translational research in PDAC. In particular, they focus on photodynamic therapy of PDAC for its ability, in addition to directly killing cancer cells, to perturb the tumor microenvironment, which is a major challenge in the treatment of this fatal tumor type. The development of ex vivo PDAC models able to mimic the complexity and heterogeneity of its microenvironment is crucial for properly appraising anti-PDAC therapeutics in a preclinical setting.

General comments

Overall, this review is well written, well organized, and clear. Furthermore, it covers an interesting and promising topic, i.e. the potential of photodynamic therapy (PDT) on the treatment of PDAC and how to improve it. A proper use of figures and tables further aids understanding of the topic.

Chapter 2, “Two-dimensional and three-dimensional PDAC models” discusses in detail the main 3D models available in the preclinical setting for PDAC studies, highlighting their advantages and limitations.

Chapter 3, “Evolving Technologies for PDAC Mimicry” is more of a concluding paragraph in Chapter 2 than a chapter in itself. It is suggested to review chapters organization. Consistent with the discussion carried out in this paragraph, I would suggest the author to have a look to PMID: 37174038, in which a similar analysis has been performed ,highlighting the effect of another key feature of PDAC TME, i.e. acidosis, in the transcriptional landscape of PDAC. Of note, I am one of the authors of this paper, and I am not suggesting this paper for any other reason than a scientific matter; therefore, I invite authors to check the paper and to add the citation only if they consider it suitable and appropriate to their discussion.

Specific comments

The resolution of the figures could be improved.

Review the formatting of Tables because some inappropriate line breaks make it more confusing to read.

Line 114: “as well as”

Lines 234-235: the author state that “Lower cell viability was seen following treatment with 100 μM of free gemcitabine in the co-cultured spheroids compared to the mono-cultured spheroids, demonstrating a higher level of drug resistance in the co-cultures”; however the numbers reported in parenthesis show the opposite (67% vs 50%) and, moreover, lower viability indicates higher chemosensitivity and lower chemoresistance, not the opposite. Please check this statement.

Lines 643-644: “Another important biological factor that is often dismissed in local biochemical milieu” the verb is missing

Line 898: “results”

Lines 1116-1118: ”It should be noted that the first two points only apply to  spheroids and organoids since PDXs and other forms of ex vivo models with tumour tissue” incomplete sentence

Check some extra spaces within the text (for example, lines 530, 791, 853, etc)

Round 2

Reviewer 1 Report

Comments and Suggestions for Authors

Accept.